# Localisation-to-delocalisation transition of moiré excitons in WSe$_2$/MoSe$_2$ heterostructures

Elena Blundo [1] ✉, Federico Tuzi[1], Salvatore Cianci [1], Marzia Cuccu[1], Katarzyna Olkowska-Pucko[2], Łucja Kipczak [2], Giorgio Contestabile[1], Antonio Miriametro[1], Marco Felici [1], Giorgio Pettinari [3], Takashi Taniguchi [4], Kenji Watanabe [5], Adam Babiński[2], Maciej R. Molas [2] & Antonio Polimeni[1] ✉

Moiré excitons (MXs) are electron-hole pairs localised by the periodic (moiré) potential forming in two-dimensional heterostructures (HSs). MXs can be exploited, e.g., for creating nanoscale-ordered quantum emitters and achieving or probing strongly correlated electronic phases at relatively high temperatures. Here, we studied the exciton properties of WSe$_2$/MoSe$_2$ HSs from $T = 6$ K to room temperature using time-resolved and continuous-wave micro-photoluminescence also under a magnetic field. The exciton dynamics and emission lineshape evolution with temperature show clear signatures that MXs de-trap from the moiré potential and turn into free interlayer excitons (IXs) for temperatures above 100 K. The MX-to-IX transition is also apparent from the exciton magnetic moment reversing its sign when the moiré potential is not capable of localising excitons at elevated temperatures. Concomitantly, the exciton formation and decay times reduce drastically. Thus, our findings establish the conditions for a truly confined nature of the exciton states in a moiré superlattice with increasing temperature and photo-generated carrier density.

Two-dimensional (2D) heterostructures (HSs) can be formed by stacking two (or more) monolayers (MLs) of different van der Waals crystals. 2D HSs offer a countless number of combinations thanks to the nearly arbitrary choice of the chemical composition of the individual constituents and the control of their relative angular alignment given by the twist angle $\theta$[1]. Inherent to the stacking process is the formation of a moiré superlattice that superimposes on the topographic and electronic structure of the single MLs. This phenomenon has been particularly investigated in HSs made of transition metal dichalcogenide (TMD) semiconductors, which feature a sizeable band gap[2–8]. The moiré potential can be as deep as 100 meV[3,4,7] and can localise both intralayer excitons (Xs) residing in the MLs of the HS[9] and interlayer excitons (IXs)[3–6] and trions[10], in which different charge carriers reside in the different layers of the HS. Moiré-confined IXs (hereafter, moiré excitons, MXs) are especially interesting as they can be exploited as nanoscale-ordered arrays of quantum emitters[3,11]. Furthermore, their space-indirect character endows IXs, and specifically MXs, with long lifetimes[4,5] that, in conjunction with the depth of the moiré potential, make them suitable for the observation of high-temperature (>100 K) Bose-Einstein condensates, as shown in a WSe$_2$/MoSe$_2$ HS[12]. The topology of the moiré potential also induces strongly correlated electron and exciton states[13,14] that led to the observation of an exciton insulator surviving up to 90 K in a WS$_2$/bilayer-WSe$_2$ HS[15]. In addition, the MXs themselves were employed as a probe of the

[1]Physics Department, Sapienza University of Rome, Piazzale Aldo Moro 5, 00185 Rome, Italy. [2]Institute of Experimental Physics, Faculty of Physics, University of Warsaw, Pasteura 5, 02-093 Warsaw, Poland. [3]Institute for Photonics and Nanotechnologies, National Research Council, 00133 Rome, Italy. [4]International Center for Materials Nanoarchitectonics, National Institute for Materials Science, 1-1 Namiki, Tsukuba 305-0044, Japan. [5]Research Center for Functional Materials, National Institute for Materials Science, 1-1 Namiki, Tsukuba 305-0044, Japan. ✉e-mail: elena.blundo@uniroma1.it; antonio.polimeni@uniroma1.it

existence of Mott insulators and Wigner crystals in WSe$_2$/WS$_2$ HSs at relatively large temperatures[16,17].

For boson condensates and highly correlated charge systems, as well as quantum photonics applications, the thermal stability of the moiré-induced confinement of the excitons plays a crucial role and a fundamental question arises: Up to what extent can MXs be regarded as truly moiré-confined?

In this work, we addressed this important aspect by investigating the effects of the lattice temperature and of the photo-generated exciton density on the localisation of MXs, as resulting from their: (i) luminescence intensity and lineshape, (ii) temporal dynamics, (iii) magnetic moments. Specifically, we studied the emission properties of two exemplary WSe$_2$/MoSe$_2$ HSs by continuous-wave (cw) micro-photoluminescence ($\mu$-PL) measurements, also under magnetic field, and by time-resolved (tr) $\mu$-PL. Low-temperature ($T = 6$ K) tr-$\mu$-PL shows that the MX signal is characterised by different spectral components, with formation and recombination dynamics indicative of the presence of a multi-level electronic potential[4,6,18,19], as well as of MXs localised at minima with different atomic registry as apparent from their gyromagnetic factor[20]. The temperature evolution of the HS emission properties presents clear signatures of IX de-trapping from the moiré potential at $T$s above 100 K and the ensuing spectral predominance of free IXs at higher temperatures. Concomitantly, Zeeman-splitting measurements reveal an unexpected sign reversal of the exciton magnetic moment taking place with the temperature-induced MX transition to a free IX regime. This transition is paralleled by a strong reduction of both the emission rise and decay times, which mirrors the faster formation and recombination dynamics, respectively, of the free IXs.

## Results

### Moiré exciton dynamics at low temperature

The two investigated WSe$_2$/MoSe$_2$ HSs were fabricated by depositing the first ML on the substrate (directly on a SiO$_2$/Si substrate in one HS, and on a h-BN flake deposited on a SiO$_2$/Si substrate in the other HS) and by then depositing the second ML on top. The HSs were then capped with a thin h-BN layer; see Methods for other details. A comparison of the PL properties before and after the h-BN capping (see Supplementary Note 1) did not show sizeable variations of the HS optical properties. Both HSs are characterised by a rather homogeneous PL signal, both in terms of lineshape and intensity, see Supplementary Note 1. The twist angle between the MoSe$_2$ and WSe$_2$ MLs was estimated to be $\theta = 0.46°$ in the first HS (hereafter HS1) and $0.74°$ in the second (hereafter HS2), as detailed next. Such angles are close to $0°$ and correspond to the configuration referred to as R-type (while HSs with $\theta$ close to $60°$ are named H-type). We will discuss mainly the results obtained on HS1 with $\theta = 0.46°$ and refer to HS2 ($\theta = 0.74°$) as a comparative case. Figure 1a shows an optical microscope image of HS1 along with its sketch. Cw and tr-$\mu$-PL measurements were carried out at variable laser excitation power $P_{\text{exc}}$ and temperature using a confocal microscope setup. For $\mu$-PL excitation ($\mu$-PLE) measurements, we employed the same setup using a wavelength-tunable laser as excitation source. Magneto-$\mu$-PL measurements were performed at variable temperature in a superconducting magnet up to 16 T, with the field perpendicular to the HS plane. Further details are reported in the Methods section.

Figure 1 b shows the $T = 6$ K $\mu$-PL spectrum (wine line) of HS1. Two bands are observed. The one peaked at 1.6 eV, labelled X, is due to a group of localised (intralayer) exciton states originating from the MoSe$_2$ ML with a small contribution from similar transitions in the WSe$_2$ ML on the higher energy side of the band. The band centred at $\approx 1.36$ eV, labelled MX, is due to MX recombination (with the electron and hole being confined in the MoSe$_2$ and WSe$_2$ layer, respectively), as also reported in other works[4,6,7,13]. The blue line in Fig. 1b is the $\mu$-PLE spectrum obtained by monitoring the MX signal while

scanning the excitation laser wavelength. The MX signal shows a resonant contribution from the MoSe$_2$ and WSe$_2$ ML exciton states of the HS, thus confirming the interlayer nature of the MX band. We point out that, at variance with ref. 21, no MX-related absorption feature is instead observed in the $\mu$-PLE data due to the much smaller oscillator strength of the MX absorption. Figure 1c displays the MX spectrum recorded at $T = 6$ K with $P_{\text{exc}} = 5$ nW (corresponding to 0.64 W/cm$^2$). The spectrum can be deconvolved into several Gaussian components. The latter are equally spaced by $(12.8 \pm 1.3)$ meV, reflecting the quantised states of the moiré potential[4,6,18,19]. The Gaussian lineshape maps onto the ensemble of MXs confined in randomly distributed moiré minima, due to the inevitable imperfections present in the HS plane. The very narrow lines superimposed on the multi-Gaussian lineshape of the MX band likely correspond to single MXs confined in just one moiré minimum[7,22]. The centroid energy of the MX band (1.357 eV) indicates that the investigated HS is R-type ($\theta \approx 0°$)[4–6,13,23] with R$_h^X$ local atomic registry[3,7]. In fact, for H-type HSs ($\theta \approx 60°$) the MX

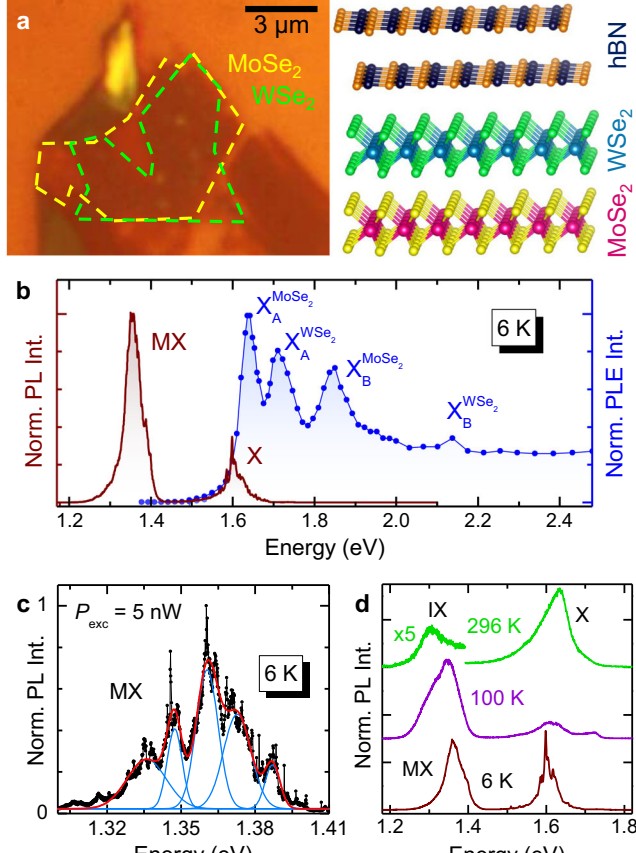

**Fig. 1 | Optical properties of the WSe$_2$/MoSe$_2$ R-type HS1. a** Optical micro-graph (left) and sketch (right) of HS1. **b** Low-$T$ $\mu$-PL and $\mu$-PLE spectra of the HS, left and right axis, respectively. In the $\mu$-PL spectrum ($P_{\text{exc}} = 2\,\mu$W), X indicates the intralayer exciton recombination from localised states of the MoSe$_2$ and WSe$_2$ monolayers (lower- and higher-energy side, respectively). MX is the moiré exciton band. In the $\mu$-PLE spectrum, four exciton resonances are observed. These resonances can be attributed to the A and B excitons (where the hole sits in the upper, A, and lower, B, spin-split valence band maximum at K, and the electron sits in the spin-split conduction band minimum at K with same spin) of the MoSe$_2$ and WSe$_2$ layers. **c** $\mu$-PL spectrum of the MX band acquired with very low laser power excitation (5 nW). The spectrum can be reproduced by five Gaussian functions (azure: single components; red line: total fit) that are spaced by $(12.8 \pm 1.3)$ meV. The very narrow lines that make up the broader Gaussian peaks correspond to single MXs recombining in moiré minima. **d** $\mu$-PL spectra recorded at different temperatures (and $P_{\text{exc}} = 20$ $\mu$W). The moiré/interlayer (MX/IX) exciton band is visible up to room temperature. X indicates the exciton band related to the single layer MoSe$_2$ and WSe$_2$ constituents of the HS.

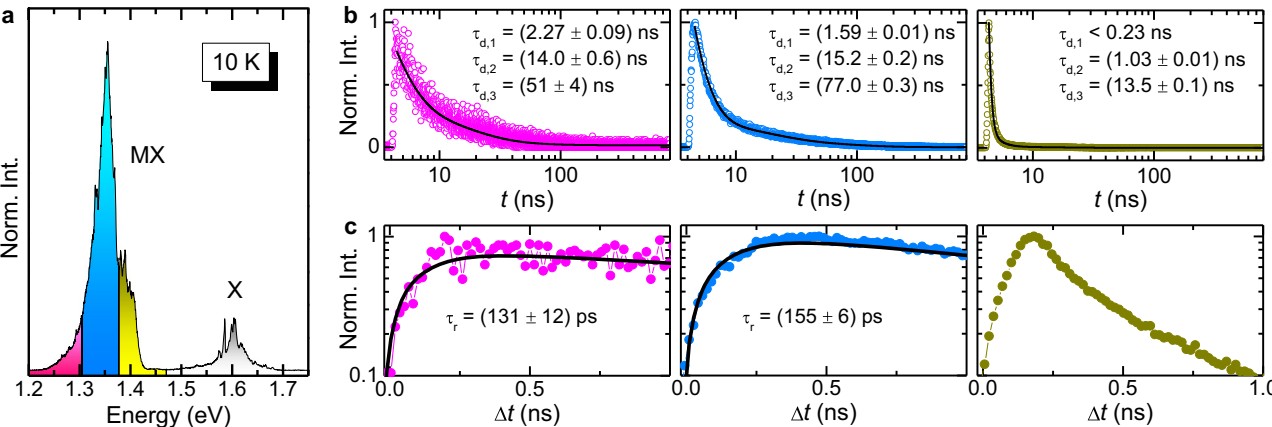

**Fig. 2 | Decay and rise of the moiré exciton band. a** $T = 10$ K and $P_{exc} = 1\,\mu$W $\mu$-PL spectrum of HS1. MX indicates the moiré-trapped interlayer exciton, and X indicates the intralayer exciton recombination. Three different spectral regions are highlighted on the MX band. On each of these regions, the $\mu$-PL time evolution was recorded. **b** Time-evolution of the $\mu$-PL signal recorded in the $\Delta t = 0$-800 ns interval from the laser pulse on the three spectral regions highlighted in panel **a** (note also the colour code). The decay time $\tau_{d,n}$ values obtained by fitting the data via Eq. (1) (see solid lines) are displayed. **c** The same as **b** for $\Delta t = 0$-1.0 ns. The rise time $\tau_r$ values displayed in the panels are those used to reproduce the data with Eq. (2) (see solid lines). The data in the right-most panel could not be fitted reliably.

recombination is centred at a higher energy −by about 40 meV[3,5,7,10,11,22,24−31]− due to the shallower moiré potential for H-type HSs with respect to R-type HSs[7], see also Supplementary Note 1. From the spacing between the MX states, as detailed in Supplementary Note 2, we estimate a moiré superlattice period $a_m$ of about 40 nm, which corresponds to $\theta = \left(0.46^{+0.05}_{-0.04}\right)^{\circ}$[10]. Second harmonic generation (SHG) measurements confirm that estimation and provide $\theta = 0.1^{\circ} \pm 1.5^{\circ}$, as reported in Supplementary Note 2. Given the large uncertainty of the SHG data, we will assign to this HS the twist angle $\theta = 0.46^{\circ}$, determined by the energy spacing of the moiré potential resonances. From the HS period, we deduce that about 600 moiré minima are probed within the laser spot (radius equal to $\approx 500$ nm). The excellent alignment leads to a sizeable signal of the HS IXs up to room temperature, as shown in Fig. 1d. Note that the recombination from the HS is indicated as MX at $T = 6$ K and as IX at $T = 296$ K, qualitatively hinting at a temperature-induced transition in the character of the exciton. We investigated such transition by studying the temporal evolution of the HS exciton signal, its dependence on the number of photo-generated carriers and by determining the exciton gyromagnetic factor at different temperatures and photo-generated carrier densities.

We first describe the tr-$\mu$-PL results at $T = 10$ K, where most of the HS emission is due to the MX recombination in the $R^X_h$ minima of the moiré potential[3,7,32]. Figure 2a shows the $\mu$-PL spectrum of HS1 recorded at a power 200 times larger ($P_{exc} = 1\,\mu$W, i.e., 128 W/cm$^2$) than in Fig. 1c. This results in a non negligible contribution from a component centred at about 1.4 eV, which can be assigned (totally or partly) to moiré excitons confined in the $R^h_h$ minima (as confirmed later). Indeed, the energy distance between this contribution and the lowest energy component (1.335 eV) of the HS emission at very low power density (see Fig. 1c) is about 65 meV. This value falls within the range of the band gap energy difference between the $R^X_h$ and $R^h_h$ moiré evaluated in refs. 3 (40 meV) and ref. 4 (70 meV). Three different spectral windows are highlighted in Fig. 2a. For each of them, panel b and panel c display the corresponding $\mu$-PL signal time evolution from the laser pulse up to 800 ns and in the time interval (0-1) ns, respectively (see also Supplementary Note 3, where the same data are compared to the laser pulse setting the resolution limit of our optical system). In the former range, the decay part of the data can be fitted by

$$I_{decay}(t) = \sum_{n=1}^{3} A_{d,n} \cdot \exp\left(-\frac{t - t_0}{\tau_{d,n}}\right), \qquad (1)$$

where $t_0$ is a reference time (from which the decay starts), and $\tau_{d,n}$ is the decay time relative to the $n$-th component, whose weight is given by $w_{d,n} = A_{d,n}/(A_{d,1} + A_{d,2} + A_{d,3})$. The fitting curves are superimposed to the data as solid lines in Fig. 2b and the $\tau_{d,n}$ values are displayed in the same figure (the complete set of the fitting parameters, including $w_{d,n}$, can be found in Supplementary Note 3). The presence of different components (1, 2 and 3) indicates that different intermediate and intercommunicating levels are involved in the MX decay, possibly including dark exciton states[6,19]. In any case, $\tau_{d,n}$ gets shorter for the higher energy ranges considered; this is particularly true for the 1.4 eV component, similar to recent results[4,6,13,19]. This finding supports the hypothesis that the structured MX emission corresponds to a ladder of discrete states arising from the moiré potential[4,6,19]. Indeed, higher-energy states may decay faster (due to the tendency of photo-generated carriers to occupy lower-lying states), with the ground state having the longest lifetime of several tens of ns, consistent with the spatially and k-space indirect characteristics of the MX transition. We recall that in TMD MLs the intralayer exciton X is known to have much shorter radiative decay times, on the order of a few ps to a few of tens of ps[33,34], in contrast with MX. A similar behaviour is found in the second WSe$_2$/MoSe$_2$ HS investigated, or HS2, with energy spacing between the moiré resonances equal to $(20.3 \pm 3.4)$ meV and twist angle $\theta = (0.74^{+0.16}_{-0.11})^{\circ}$[10]. The data for HS2 can be found in Supplementary Note 2. As shown there, it is worth remarking that for HS2, featuring greater $\theta$ than HS1, longer decay times $\tau_{d,n}$ are found. This is in accordance with the larger momentum mismatch between the charge pair of the moiré exciton in HS2 and agrees with the results presented in ref. 6.

The different states of the moiré potential also exhibit a different formation dynamics. Figure 2c shows the time evolution of the MX signal in HS1 up to 1 ns after the laser pulse excitation. In this case, the data are reproduced by

$$I_{rise}(t) = -A_r \cdot \exp\left(-\frac{t - t_0}{\tau_r}\right) + I_{decay}(t) \qquad (2)$$

where $\tau_r$ is the luminescence rise time and $I_{decay}$ represents the decay part of the data. By fitting the decay part first, the data in the (0-1) ns time interval can be reproduced by Eq. 2, with only $A_r$ and $\tau_r$ as fitting parameters. The $\tau_r$ values are displayed in panel c of Fig. 2 (the data corresponding to the high-energy range, shown in the right-most panel, are close to the resolution limit and could not be fitted reliably). The data indicate that the highest-energy excited state of the moiré

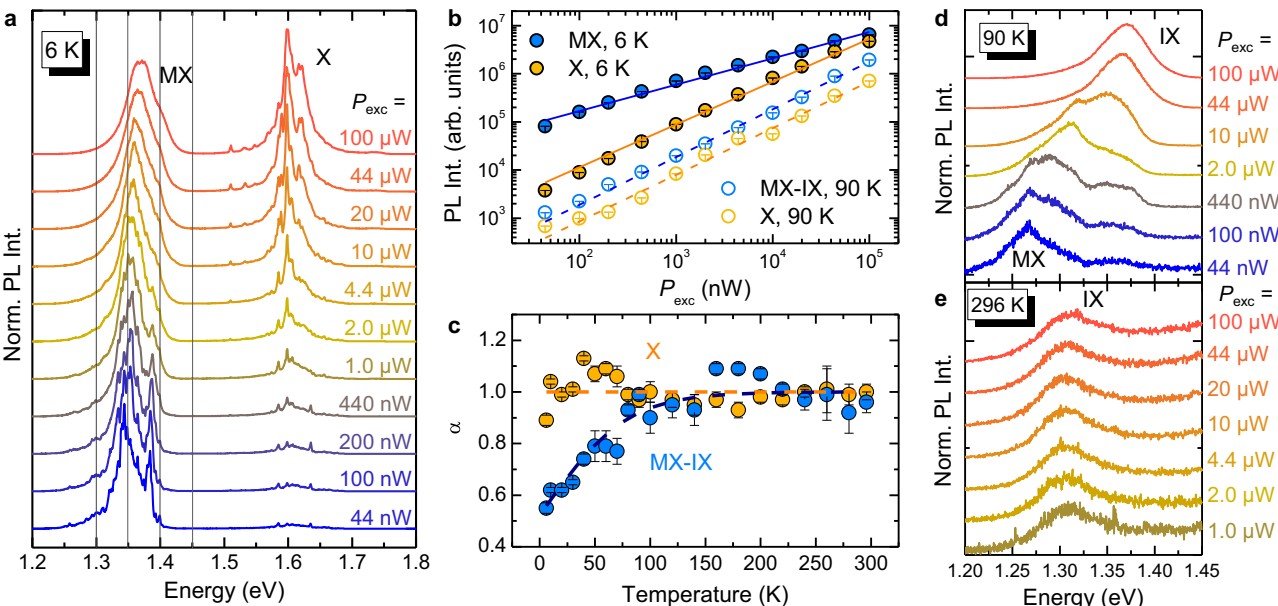

**Fig. 3 | Photo-generated carrier density and temperature dependence of the exciton bands. a** $T = 6$ K $\mu$-PL spectra of HS1 recorded for different laser excitation power values. MX indicates the moiré exciton band and X the intralayer exciton recombination in the MoSe$_2$ and WSe$_2$ layers (lower- and higher-energy side, respectively). **b** PL integrated intensity dependence on the laser power $P_{exc}$ for the MX or MX-IX bands (light orange symbols) and for the X band (azure symbols) at $T = 6$ K (full symbols) and $T = 90$ K (open symbols), respectively. Solid and dashed lines are fits to the data with Eq. (3) for $T = 6$ K and 90 K, respectively. At $T = 6$ K, the $\alpha$ coefficient values are 0.55 ± 0.02 and 0.89 ± 0.02 for MX and X, respectively.

At $T = 90$ K, the $\alpha$ coefficient values are 0.99 ± 0.02 and 0.97 ± 0.03 for MX-IX and X, respectively. **c** Temperature variation of the $\alpha$ coefficient for the MX-IX and X bands. In the former case, a clear transition from a sublinear to a linear behaviour is found and ascribed to the transition from a moiré localisation regime to a free interlayer exciton one (hence the mixed label MX-IX). **d** $T = 90$ K $\mu$-PL spectra for different laser excitation powers in the energy region where the MX and IX recombinations can be simultaneously observed. IX takes over MX upon increase of the photo-generated carrier density. **e** Same as **d** for $T = 296$ K, where only the IX transition is observable.

potential (together with the likely population of moiré excitons in the relative minimum at the R$_h^h$ registry) is populated faster (<100 ps), similar to what reported in ref. 19. Instead, the population of the lowest-energy state requires more time to reach its quasi-equilibrium occupancy because of the extra contribution from higher-energy levels in addition to the direct excitation. Power-dependent tr-$\mu$-PL measurements on both HSs, reported in Supplementary Note 4 and 5, show a progressive shortening of the time decay of the MX band as $P_{exc}$ increases. This is a likely consequence of exciton-exciton interactions, that tend to diminish the exciton lifetime[35,36].

**Exciton recombination vs carrier density and temperature**

The X and MX recombination bands also exhibit quite distinct spectral behaviours when the density of photo-generated excitons and the lattice temperature are increased. Figure 3a shows the cw $\mu$-PL spectra of HS1 at $T = 6$ K for $P_{exc}$ ranging from 44 nW (i.e. 5.6 W/cm$^2$) to 100 $\mu$W (i.e. 1.3 · 10$^4$ W/cm$^2$). The MX band broadens and its centroid blueshifts with increasing $P_{exc}$, likely as a consequence of the dipole-dipole interaction between MXs[5,13,22,26,37,38]. Following Ref. 13, we determine that in the $P_{exc} = (0.044$-$100)$ $\mu$W range the density of photo-generated electron-hole pairs within the HS varies from $n_{e\text{-}h} = 1.1 \cdot 10^{11}$ cm$^{-2}$ to $2.3 \cdot 10^{13}$ cm$^{-2}$ (see Supplementary Note 4). We note that the highest $n_{e\text{-}h}$ achieved by us is smaller than the value necessary to observe an optically induced Mott transition from IXs to spatially separated electron and hole gases[13]. Nevertheless, from the previously estimated period of the moiré potential $a_m = 40$ nm, the corresponding density of moiré minima is equal to $7.2 \cdot 10^{10}$ cm$^{-2}$ and a sizeable exciton-exciton interaction is possible thus explaining the decrease in the emission decay time reported in Supplementary Note 4 and 5 as well as the MX band blueshift with $P_{exc}$[13,22,38]. Recent simulations based on the Green's function formalism[38] showed that exciton densities around $10^{11}$ – $10^{12}$ cm$^{-2}$ trigger intercell hopping and exciton delocalisation effects, which reflect in a blue-shift and broadening of the moiré emission band. In

the present case, we observe such effects starting from $P_{exc} = 4.4$ $\mu$W (see Fig. 3b), corresponding to $n_{e\text{-}h} \approx 4 \cdot 10^{12}$ cm$^{-2}$, in good agreement with ref. 38. On the other hand, the X band, which, as we recall, comprises the MoSe$_2$ and WSe$_2$ intralayer excitons, does not change appreciably its centroid. It instead gains significant spectral weight compared to MX, which originates from recombination centres with finite spatial density. Figure 3b shows the dependence of the integrated intensity $I$ of the HS exciton and X bands as a function of $P_{exc}$ for $T = 6$ K and $T = 90$ K. At 90 K, the HS exciton band is labelled MX-IX to indicate the contribution from IXs (or de-trapped MXs) occurring at high $P_{exc}$ and increasing temperature, as we are going to demonstrate. The integrated intensity was obtained by performing integrals over suitable energy ranges (it cannot be reliably obtained by fitting the data since the shape of the PL bands changes with power, comprising the appearance of narrow lines at low power). The data were fitted by:

$$I = A \cdot P_{exc}^{\alpha}, \tag{3}$$

where $A$ is a scaling constant. At $T = 6$ K, $\alpha$ is equal to 0.55 ± 0.02 for MX and 0.89 ± 0.02 for X. The smaller $\alpha$ found at low $T$ for the MX signal from the HS (as opposed to that from intralayer excitons X in the MLs) is compatible with the finite number of energy states available for excitons trapped in the quantised levels of the moiré potential minima. Furthermore, the localised nature of MXs can lead to enhanced exciton-exciton interactions that act as a probable source of signal loss. Instead, the nearly linear behaviour of the X emission intensity is consistent with the virtually unlimited number of intralayer excitons that can be photo-generated. Interestingly, Fig. 3b shows that the nearly linear dependence of the X band on $P_{exc}$ is maintained also at $T = 90$ K, while a major variation is found for the MX-IX band that can be explained by considering an increasingly higher spectral contribution of free IXs at higher $T$. As a matter of fact, the $\alpha$ value of MX-IX becomes approximately equal to 1 at 90 K.

Puzzled by this finding, we investigated the dependencies on $P_{exc}$ of the integrated area of the MX-IX and X bands at different temperatures. The full set of power-dependent data can be found in Supplementary Note 6. Figure 3c summarises the variation of the coefficient $\alpha$ with $T$, as obtained from Eq. (3). For the X band, a nearly linear behaviour is observed at all temperatures. Instead, for the MX-IX band, $\alpha$ increases progressively from 0.55 to about 1 as $T$ is increased from 5 K to 120 K, with a linear behaviour observed at higher temperatures, up to room temperature. These results suggest a qualitative change in the nature of the exciton-related bands in the HS at about 120 K. A similarly comprehensive study was performed also for HS2, which has a larger twist angle of 0.74°. The results are shown in Supplementary Note 7. As for HS1, the X band shows a linear behavior ($\alpha$ = 1) all over the temperature range. Although also in HS2 the coefficient $\alpha$ for the MX-IX band exhibits a progressive increase from 0.5 to 1 (see Supplementary Note 7), the plateau-value of 1 is reached at a temperature of about 100 K (lower than the 120 K value observed in HS1; see Fig. 3c) consistently with the shallower moiré potential at greater twist angles[39]. On the one hand, these results strengthen the picture described so far. On the other hand, they point to the crucial role of the twist angle in determining the thermal stability of moiré excitons and associated phenomena in moiré superlattices.

Figures 3d and e display a series of spectra recorded on HS1 at different $P_{exc}$ for $T$ = 90 K and $T$ = 296 K, respectively. In the first case, the lineshape of the MX band changes significantly as the density of photoexcited carriers increases. Indeed, we notice a considerable spectral weight transfer by about 100 meV from the structured band around 1.27 eV (at the lowest $P_{exc}$) to the single component peaked at about 1.37 eV (at the highest $P_{exc}$). We ascribe this change to the saturation of moiré-localised excitons in favour of moiré-de-trapped IXs (the related 100 meV band shift is indeed close to the moiré potential depth for R-type WSe$_2$/MoSe$_2$ HS[3,4,7]). This behaviour is not evident at the lowest $T$ values (see, e.g., Fig. 3a), when MXs are frozen in their potential minima. Eventually, for $T$ > 200 K, almost all MXs are ionised and only IXs are observed, as shown in Fig. 3e, clearly demonstrating the absence of a sizeable lineshape variation with $P_{exc}$. See Supplementary Note 7 for similar data recorded on HS2 (having greater $\theta$).

## Moiré exciton de-trapping, magnetic moment and dynamics vs $T$

The moiré exciton de-trapping is even more evident in Fig. 4a, which shows the $\mu$-PL spectra of HS1 for different $T$ values and $P_{exc}$ = 10 $\mu$W; similar studies for higher and lower $P_{exc}$ are shown in Supplementary Note 8. From $T$ = 6 K to $T$ = 120 K, the HS signal is dominated by the MX band, which undergoes a redistribution of the carrier population between the different states of the moiré potential. Starting from $T$ = 120 K, a high-energy component due to IXs appears and becomes increasingly important relative to the MX band, until the latter vanishes at about 220 K. Finally, at room temperature only IXs are visible. At $T \approx 160$ K, the two contributions coexist so that their energy difference can be estimated. The obtained value, equal to about 90 meV, fits well with the exciton barrier height of the moiré potential in R-type WSe$_2$/MoSe$_2$ HSs[3,4,7], where only the exciton singlet state is optically permitted (the MX-IX energy distance becomes 75 meV in the HS2 with greater $\theta$). Although qualitatively, also this observation agrees with the expected shallower moiré potential depth for greater twist angles[39]; see Supplementary Notes 6 and 7. In contrast, the exciton ground state in H-type HSs is in a triplet configuration, with the singlet state having an energy 25 meV higher[5,22,27]. Also, we exclude that the two transitions coexisting at intermediate $T$ are ascribable to $K_{CB}$-$K_{VB}$ (CB and VB stand for conduction and valence band, respectively) and $\Lambda_{CB}$-$K_{VB}$ IX transitions[21], which differ by 55 meV[21].

It is worth mentioning that different results were reported in the literature. In ref. 28, the MX de-trapping was observed by monitoring the PL intensity and lifetime of WSe$_2$/MoSe$_2$ HSs with a transition temperature < 50 K that is in contrast with our results. On the other hand, exciton diffusivity measurements[7] showed the absence of MX de-trapping in a WSe$_2$/MoSe$_2$ HS with nearly perfect lattice alignment ($\theta$ = 0.15°), while a clear de-trapping was visible for $\theta > 2$°[7].

In any case, the here observed temperature-induced change in the nature of the exciton in the HS should be reflected in the electronic properties of the levels involved in the excitonic transition. In this respect, the exciton magnetic moment and the associated gyromagnetic factor $g_{exc}$ —embedding the spin, orbital and valley properties of the bands— turned out to be an extremely sensitive parameter of the electronic structure of nanostructures[40] and of 2D crystals[41–44] and their HSs[5,10,20,22–24,26,45–48]. In WSe$_2$/MoSe$_2$ HSs, the lowest-energy exciton state is in a spin-singlet configuration for R-type HSs and in a spin-triplet configuration for H-type HSs[49]. The spin-singlet and spin-triplet excitons feature a $g_{exc}$ value with a positive ($\approx$+ 7) and a negative ($\approx$−15) sign, respectively, the exact value depending on the specific sample[5,20,22,24,25,46–48]. Our HS is R-type, as discussed before, and therefore we expect a positive $g_{exc}$ value. Figure 4b shows the magnetic field, $B$, dependent $\mu$-PL spectra of HS1 in the HS exciton region at $T$ = 10 K and $P_{exc}$ = 10 nW (corresponding to $n_{e-h}$ = 2.0 · 10$^{10}$ cm$^{-2}$, see Supplementary Note 4). For each field, the opposite circular polarisations $\sigma^{\pm}$ were recorded simultaneously on two different regions of the CCD detector, see Methods. Figure 4b shows the $\sigma^+$ and $\sigma^-$-polarised $\mu$-PL spectra that exhibit several narrow lines due to different MXs. They all undergo a Zeeman splitting (ZS) given by

$$ZS(B) = E^{\sigma^+} - E^{\sigma^-} = g_{exc} \cdot \mu_B B. \qquad (4)$$

$E^{\sigma^{\pm}}$ are the peak energies of components with opposite helicity $\sigma^+$ and $\sigma^-$, and $\mu_B$ is the Bohr magneton. The positive (negative) energy shift with $B$ of the $\sigma^+$ ($\sigma^-$) component of the lines displayed in Fig. 4b indicates that $g_{exc} > 0$ for the individual MXs. Then, magneto-$\mu$-PL measurements were performed also at $T$ = 160 K and $P_{exc}$=75 $\mu$W, where the HS exciton band is instead dominated by free IXs. Figure 4c shows the $\sigma^+$ and $\sigma^-$ components of the IX spectra at different magnetic fields. Remarkably, the two components shift with $B$ accordingly to a negative ZS, i.e. opposite to that found at $T$ = 10 K for the MX lines (for IXs, the $\sigma^+$ red component is at lower energy than the $\sigma^-$ blue one). Figure 4d shows the ZS field dependence for the MX lines at 10 K and for the IX band at 160 K, both fitted by Eq. (4). The resulting $g_{exc}$ for the (trapped) MXs and (free) IXs are $g_{exc,MX}$ = + 6.73 ± 0.10 (this is an average value over the 5 measured MXs) and $g_{exc,IX}$ = − 4.64 ± 0.10, respectively. The former is in close agreement with previous experimental[5,23,46,47] and theoretical[20] results found for MXs in R-type WSe$_2$/MoSe$_2$ HSs. As reported in Supplementary Note 9, we performed similar measurements and found similar results for HS2. For HS1, we also derived the ZS of the five gaussians into which the moiré exciton band can be deconvolved at low $T$ and $P_{exc}$, as shown in Fig. 1c. As discussed in Supplementary Note 10, in comparison to the single MX lines of Fig. 4d, a smaller ZS is found for the Gaussian components, resulting in an average $g_{exc}$ = + 4.43 ± 0.89. The smaller $g_{exc}$ for the Gaussian components might be caused by exciton-exciton interactions. In fact, the Gaussian lineshape could be the consequence not only of a distribution of an ensemble of single moiré lines, but also of an exciton interaction-induced broadening of the the moiré emission itself. As for the results at 160 K, to our knowledge there are no previous ZS measurements on HSs at high temperatures. We found a similar negative $g_{exc,IX}$ value of about −5 also at $T$ = 210 K and 100 K, as described in Supplementary Note 11. The origin of the sign reversal of $g_{exc,IX}$ must be then ascribed to the avoided effect of the moiré potential caused by the temperature-induced de-trapping of the MXs. As a matter of fact, we can estimate $g_{exc,IX}$ considering the separate contribution of electrons and holes to the IX gyromagnetic factor, as usually done for excitons in semiconductors. Following an analogous

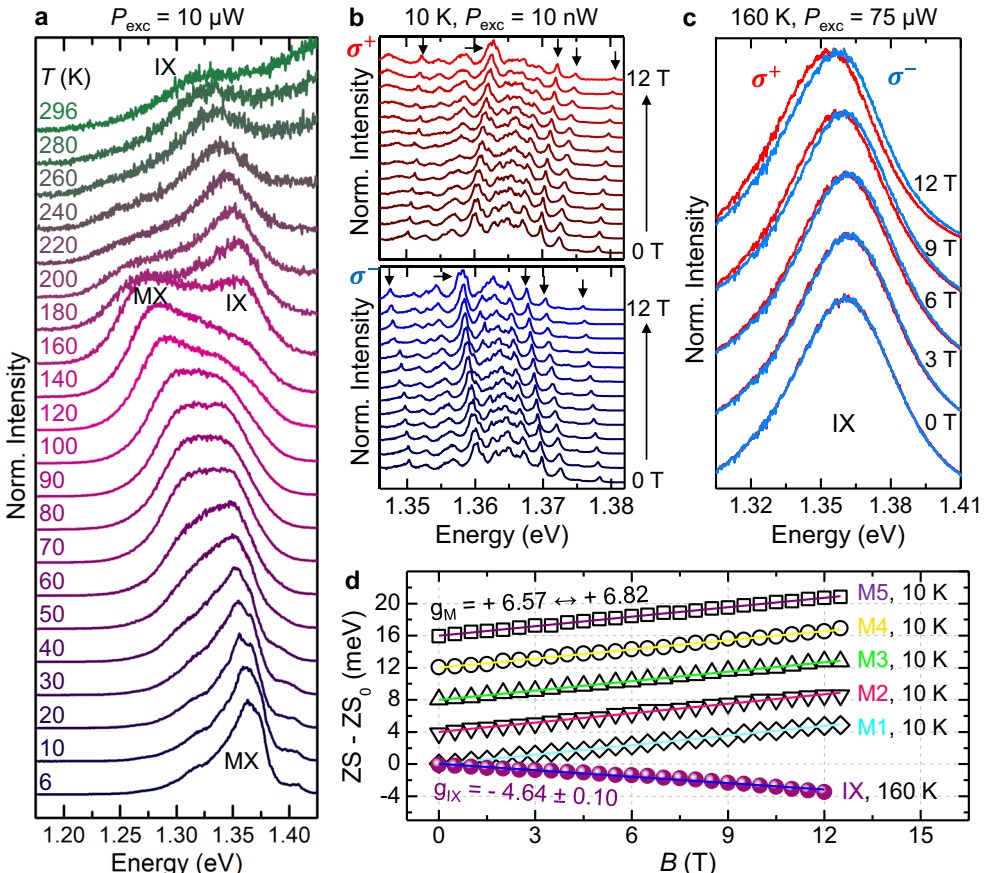

**Fig. 4 | Exciton magnetic moment sign reversal. a** $\mu$-PL spectra of HS1 recorded for different temperatures and fixed $P_{exc} = 10$ $\mu$W focused via a $20 \times$ objective (NA=0.4). MX indicates the moiré exciton band and IX the free interlayer exciton recombination. Note the major spectral transfer from MXs to IXs for $T > 120$ K. **b** Magneto-$\mu$-PL spectra from 0 to 12 T (in steps of 1 T) of the MX band at $T = 10$ K and $P_{exc} = 10$ nW (focused via a 100× objective with NA = 0.82). The upper and lower panels correspond to the $\sigma^+$ (red) and $\sigma^-$ (blue) circular polarisations, respectively. The data are stacked by y-offset. The positive and negative slopes of the $\sigma^+$ and $\sigma^-$ polarisations with the field indicate a positive gyromagnetic factor. The arrows denote some specific MX lines. **c** Magneto- $\mu$-PL spectra at $T = 160$ K and $P_{exc} = 75$ $\mu$W of the free IX band for the $\sigma^+$ and $\sigma^-$ polarisations. A negative Zeeman splitting (ZS) can be observed, with the $\sigma^+$ and $\sigma^-$ spectra being at lower and higher energy, respectively. **d** ZS of the five moiré-localised excitons M1 to M5 highlighted by the arrows in panel **b** (M1 is the lowest energy one, M5 the highest) and of the free IX exciton vs magnetic field measured in panel **c**, resulting in the gyromagnetic factors displayed in the figure. The ZS data of the M2 to M5 lines were shifted by y-offset for ease of visualisation. Error bars are within the symbol size.

procedure to that employed in ref. [43] for strained WS$_2$ MLs, for this HS we use

$$g_{exc,IX} = 2\left[L_{CB}(\text{MoSe}_2) - L_{VB}(\text{WSe}_2)\right], \qquad (5)$$

where the first and second terms are the expectation values of the orbital angular momentum of the MoSe$_2$ CB and WSe$_2$ VB, respectively (the spin contribution cancels out because the band extrema involved in the free IX transition have the same spin for R-type HSs). As reported in ref. [20], $L_{CB}(\text{MoSe}_2) = 1.78$ and $L_{VB}(\text{WSe}_2) = 4.00$ and from Eq. (5) we obtain $g_{exc,IX} = -4.44$ in very good agreement with the value we found experimentally for the free IX transition shown in Fig. 4d. To this regard, it is relevant to add that a positive $g_{exc}$ is found for the MX band also at $T = 80$ K, as shown by $\mu$-PL measurements on HS1 performed at $B = 16$ T and sufficiently low laser power (as low as 10 nW) to emphasize the contribution of the MX band at high $T$; see Supplementary Note 11.

Interestingly, the suppression of the moiré potential in an R-type WSe$_2$/MoSe$_2$ HS caused by inserting a h-BN layer between the constituent MLs leads to magneto-PL results similar to ours[50]. Indeed, in Ref. [50] the spatial decoupling between the HS single MLs determines a sign reversal and a decrease of the $g_{exc}$ modulus analogous to that found here by increasing the lattice temperature.

The exciton ZS in WSe$_2$/MoSe$_2$ HSs may depend on the local atomic registry, too[20]. Figure 5a shows a series of $\mu$-PL spectra recorded on HS1 with opposite circular polarisation ($\sigma^+$ and $\sigma^-$) at different magnetic fields for $T = 6$ K and low laser excitation power $P_{exc} = 0.2$ $\mu$W. Several narrow lines due to moiré confined excitons can be observed, superimposed on a continuum background. Like in Fig. 4b, the MX lines exhibit a positive Zeeman splitting with average $g_{exc,MX} = +6.78 \pm 0.11$ (panel c of the figure). Figure 5b shows the same study of panel a recorded on the same point of the HS but with a $P_{exc}$ value increased by about a factor of 400. The narrow lines associated to moiré confined excitons merge forming a single band, whose maximum shows a blue-shift of more than 30 meV with respect to the low-energy side of the MX band at lower $P_{exc}$. Quite interestingly and unexpectedly, under these circumstances, the ZS value reverses its sign resulting in a gyromagnetic factor $g_{exc} = -6.93$, as shown in Fig. 5c. We also performed a PL study at fixed $B = 16$ T by varying $P_{exc}$ over more than three orders of magnitude. The data are described in Supplementary Note 12 for both HS1 and HS2 and show how when increasing the density of excitons the initially positive ZS due to the single moiré excitons eventually goes negative as the MX component broadens and blue-shifts. To interpret these results we note that: (i) the $P_{exc}$-induced blue-shift is significantly smaller than 100 meV (i.e., the moiré potential depth[3,4,7]) and (ii) the modulus of $g_{exc}$ ($|-6.93|$) found at high $P_{exc}$ and $T = 10$ K is larger than that ($|-4.64|$)

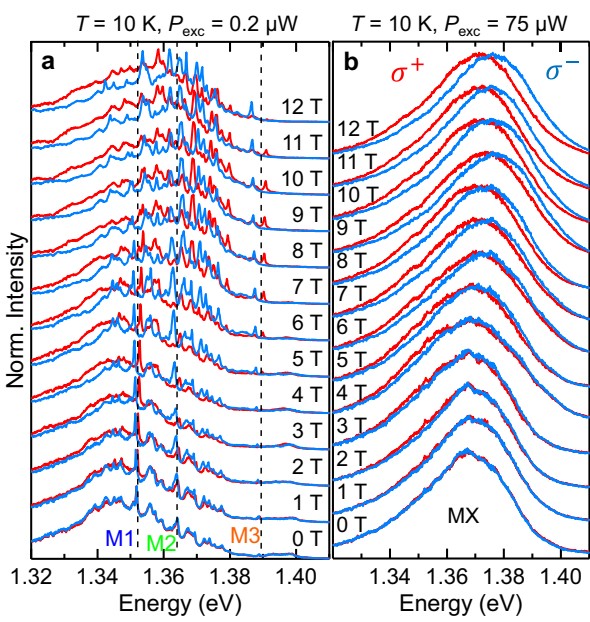

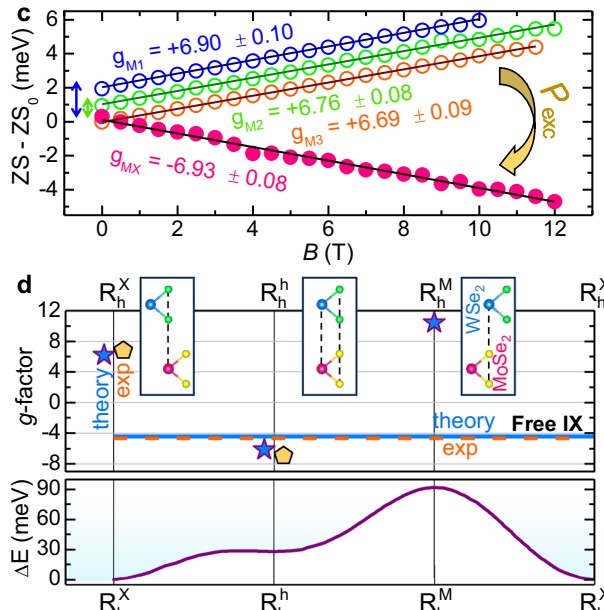

**Fig. 5 | Unveiling the moiré atomic registry through $g$-factor measurements. a, b** Helicity-resolved normalised $\mu$-PL spectra of HS1 under magnetic field at $T = 6$ K for two different laser excitation powers $P_{exc}$ (focused via a 100 × objective with NA = 0.82). The two sets of data were acquired in the same point of the HS. For $P_{exc} = 0.2$ $\mu$W (**a**), many narrow lines can be seen. M1, M2 and M3 indicate three such narrow lines. At high powers $P_{exc} = 75$ $\mu$W (**b**) a continuous band can be seen. **c** ZS of the lines M1-M3 of panel **a** and of the MX band of panel **b**, showing an opposite sign of the $g$-factor. The data of lines M2 and M3 are up-shifted by 1 and 2 meV, respectively (as indicated by the double-sided arrows on the left) for sake of clarity. **d** Top: Theoretical $g$-factors (cyan stars) estimated for MXs confined in the $R_h^X$, $R_h^h$, and $R_h^M$

atomic registries[20]. The experimental $g$-factors calculated for the MX lines at low power (as an average of the $g$-factors estimated for the M1, M2 and M3 lines of panel **c**), and for the MX band measured at high power (pink data in panel **c**) are displayed as orange pentagons, and –based on their value– are associated to the $R_h^X$ and $R_h^h$ registries, respectively. The cyan solid line and the orange dashed line provide the $g$-factor calculated based on Eq. (5) and that measured experimentally (Fig. 4c, d) for the free IX, respectively. The insets show the atomic alignment corresponding to the three atomic registries. Error bars are within the symbol/line size. Bottom: Variation of the interlayer exciton potential landscape with respect to its minimum at the $R_h^X$ point ($\Delta E$) in R-type MoSe$_2$/WSe$_2$ HSs. Adapted from the calculations of ref. 3.

measured at high $T$ (see Fig. 4c, d), where moiré de-trapped (and thus free) excitons (IXs) dominate the emission spectrum. Thus, we conclude that the main contribution to the MX band in Fig. 5b (high $P_{exc}$) comes from moiré excitons with $R_h^h$ local registry. Indeed, at low power, the lowest energy $R_h^X$ states are first populated. When $P_{exc}$ is increased, the $R_h^h$ states, that lie just a few tens of meV at higher energy (see Fig. 5d), start to be populated and, for sufficiently high power, dominate the emission spectrum. The $R_h^M$ states, that lie at even higher energy, are not visible due to their small oscillator strength[20]. This picture is corroborated by theoretical calculations, showing that $g_{exc} = +6.19$ for the $R_h^X$ registry, while $g_{exc} = -6.15$ for the $R_h^h$ registry[20]. Figure 5d summarises these findings and provide a comparison between experimental and theoretical $g_{exc}$ values for moiré-localised as well as free IX excitons. The quantitative potential landscape for the moiré potential is also shown, as derived from ref. 3.

These results demonstrate how magnetic fields represent a valuable tool to determine the localised or delocalised status of charge carriers in 2D HSs, which can be important for the understanding of fundamental effects, such as the formation of highly correlated electronic phases[12,13,16,51].

The sign reversal of the $g$-factor upon temperature increase is accompanied by a change in the MX formation and recombination dynamics when the de-trapping process starts to occur with increasing $T$. Figure 6a displays the time evolution of the $\mu$-PL exciton signal in HS1 within 1 ns after the laser excitation, which corresponds predominantly to the exciton formation process. Different temperatures were considered with $P_{exc} = 44$ nW ($n_{e-h} = 1.1 \cdot 10^{11}$ cm$^{-2}$).

It is clear that the MX formation dynamics becomes faster with increasing $T$ (for $T \leq 100$ K and $P_{exc} = 44$ nW, MXs dominate). The experimental data were fitted by Eq. (2) and the $T$ dependence of $\tau_r$ is displayed in panel **b** for two different photo-generated carrier

densities. At $T = 100$ K, $\tau_r$ approaches the temporal resolution limit (notice that once the data get close to the resolution limit, the estimated rise time is affected by the system response and thus only qualitatively indicative). The higher temperatures and the ensuing MX ionisation process result indeed in a decreased contribution of the moiré localisation step, and thus in a reduction of the time required to build up the exciton population contributing to the MX/IX band. This process is more evident with increasing $P_{exc}$, as can be noted in Fig. 6b. As a matter of fact, a larger photo-generated carrier density tends to saturate the MX states shifting the spectral centroid of the MX-IX band towards the faster-forming IX levels.

The luminescence decay is also highly influenced by temperature variations. Figure 6c shows the MX-IX band decay curves at $P_{exc} = 1$ $\mu$W ($n_{e-h} = 1.2 \cdot 10^{12}$ cm$^{-2}$) and different temperatures. The curves can be reproduced using Eq. (1) and the values of the fitting parameters ($\tau_{d,n}$ and $w_{d,n}$) are displayed in Fig. 6d. The three values of the decay time $\tau_{d,n}$ decrease monotonically, with the shorter one ($\tau_{d,1}$) reaching the resolution limit (0.23 ns) at $T = 140$ K, and the weights of the slower components ($w_{d,2}$ and $w_{d,3}$) becoming less important. The latter are particularly relevant at low $T$, where decay time values of about 200 ns are observed, consistent with the space-indirect nature of MXs. The, yet small, k-space mismatch associated with the twist angle may also contribute to the lengthening of the luminescence decay time[6]. The marked decrease of $\tau_{d,n}$ with $T$ can be explained by two simultaneous mechanisms. First, non-radiative recombination channels are activated at higher temperatures, greatly shortening the luminescence decay time. Second, delocalised states are expected to have a larger recombination probability, because they are more likely to interact with other oppositely charged free carriers, or with lattice defects acting as non-radiative channels[52].

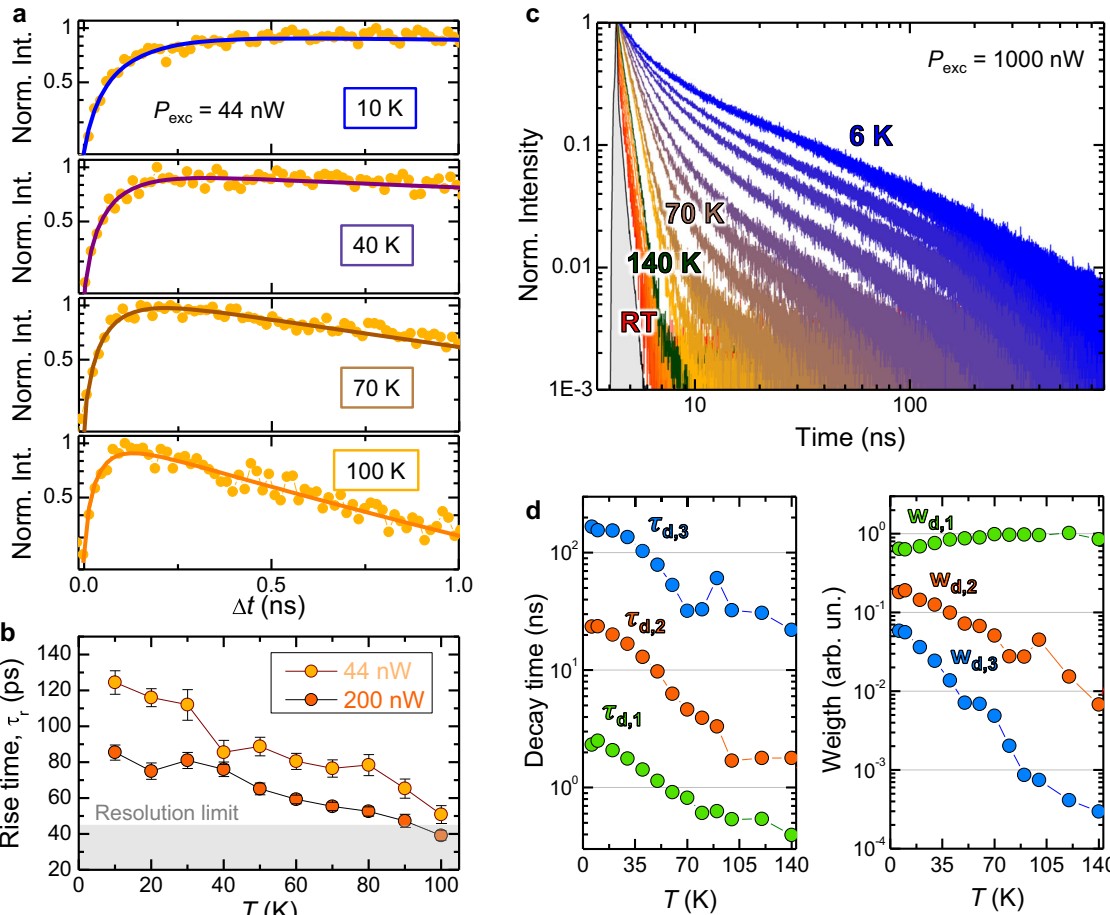

**Fig. 6 | Rise and decay times with increasing temperature. a** Time-evolution of the $\mu$-PL signal of the investigated WSe$_2$/MoSe$_2$ heterostructure (HS1) recorded at different temperatures (and fixed laser excitation power $P_{exc}$ = 44 nW) in the $\Delta t$ = 0 – 1.0 ns interval from the laser pulse. The detection energy was set at the MX-IX band (see Fig. 3). The solid lines are fits to the data by Eq. (2). **b** Rise time $\tau_r$ values obtained by fitting the experimental data for different temperatures and two $P_{exc}$ values. The setup time resolution is shown by the grey area. Notice that once the

data get close to the resolution limit, the estimated rise time is affected by the system response and thus only qualitatively indicative. **c** Time-evolution of the $\mu$-PL signal of the MX-IX band (see Fig. 3) recorded in the $\Delta t$ = 0-800 ns interval from the laser pulse. The data were recorded at different temperatures, as indicated in the figure, and fixed $P_{exc}$. The grey area indicates the instrumental response. **d** Decay times $\tau_{d,n}$ values used to reproduce the data of panel **c** via Eq. (1). The same for the spectral weights $w_{d,n}$ of the different time components, see Eq. (1).

In summary, we investigated the process of temperature-induced exciton de-trapping from moiré minima in WSe$_2$/MoSe$_2$ HSs. We observed that at temperatures above 100 K moiré excitons turn into free interlayer excitons with relevant consequences for quantum technology applications[11] and for the observation of many-body phenomena, such as exciton condensation[12] or Mott transition[13,16,51]. The temperature-induced transition from a moiré-confined to a free IX regime manifests itself in a sizeable variation of the power law governing the exciton signal growth with photo-generated carrier density. The exciton magnetic moment too undergoes major variations with increasing $T$. Indeed, the interlayer exciton $g$-factor exhibits a remarkable reversal of its sign and decrease of its modulus (going from about + 7 to about − 5) concomitantly with the de-trapping of the moiré-confined excitons for $T \gtrsim 100$ K. This is likely to have relevant consequences for valleytronic applications of TMD HSs. Within the same $T$ interval, we also consistently found that the formation time of MXs is strongly reduced, as a consequence of the cross-over from a localised to a free-like regime. This indicates that the exciton capture in the moiré potential requires an intermediate step that lengthens the luminescence rise time. Also, the decay time of the MX/IX states is greatly reduced by increasing $T$ due to the increased recombination probability of freely moving excitons as well as to exciton-exciton interactions and to thermally activated non-radiative recombination

channels. Our findings shed new light on the truly confined nature of the exciton states in a moiré superlattice with increasing temperature and exciton density, thus setting the conditions for the observation and stability of highly correlated phases at elevated temperatures in moiré superlattices.

## Methods

### Sample fabrication

The HSs were fabricated by the standard dry transfer technique. TMD flakes were mechanically exfoliated by the scotch tape method and deposited on PDMS. MoSe$_2$ and WSe$_2$ MLs on the PDMS were identified and deposited. For HS1, the MoSe$_2$ ML was deposited first on a SiO$_2$/Si substrate, and the WSe$_2$ was deposited atop of it. h-BN flakes were then exfoliated with the same approach and a thin h-BN flake was identified on the PDMS. The flake was then deposited in such a way to cap the HS completely. The twist angle between the MLs was then estimated by SHG measurements. For HS2, a h-BN flake was first deposited on a SiO$_2$/Si substrate. The WSe$_2$ ML was then deposited atop. The orientation of the ML was checked by SHG. The orientation of the MoSe$_2$ ML on PDMS was also checked, and the ML was deposited aligned to the WSe$_2$ ML (virtually null twist angle). This HS was also subsequently capped with a thin h-BN flake. The twist angle of the two HSs was then determined more precisely by optical studies (see Supplementary Note 2).

After every deposition step, the samples were annealed in high vacuum at 150 °C for some hours.

## Continuous-wave $\mu$-PL measurements

For $\mu$-PL measurements, the excitation laser was provided by a single frequency Nd:YVO$_4$ lasers (DPSS series by Lasos) emitting at 532 nm. The luminescence signal was spectrally dispersed by a 20.3 cm focal length Isoplane 160 monochromator (Princeton Instruments) equipped with a 150 grooves/mm and a 300 grooves/mm grating and detected by a back-illuminated N$_2$-cooled Si CCD camera (100BRX by Princeton Instruments). The laser light was filtered out by a very sharp long-pass Razor edge filter (Semrock). A 100 × long-working-distance Zeiss objective with NA = 0.75 was employed to excite and collect the light, in a backscattering configuration using a confocal setup.

For high resolution measurements aimed at higlighting the moiré energy levels (Fig. 1c and Supplementary Fig. 2.1), a 75 cm focal length Acton monochromator was used.

## Time-resolved $\mu$-PL measurements

For tr $\mu$-PL measurements, the sample was excited with a ps super-continuum laser (NKT Photonics) tuned at 530 nm, with a full width at half maximum of about 10 nm and 50 ps pulses at 1.2 MHz repetition rate. The sample was excited in the same experimental configuration used for continuous wave measurements. The signal was then collected in a backscattering configuration using a confocal setup. The desired spectral region was selected by using longpass and shortpass filters. The signal was thus focused by means of a lens on an avalanche photodetector from MPD with temporal resolution of 30 ps.

## $\mu$-PL excitation measurements

For $\mu$-PL excitation ($\mu$-PLE), we employed the same ps supercontinuum laser used for tr $\mu$-PL. The laser wavelength was automatically changed by an acousto-optic tunable filter and employing a series of shortpass and longpass filters to remove spurious signals from the laser. The detection wavelength was selected using the same monochromator and detector employed for cw $\mu$-PL measurements.

## Magneto-$\mu$-PL measurements

Magneto-$\mu$-PL measurements were performed at variable temperature in a superconducting magnet reaching up to 16 T. x-y-z piezoelectric stages were used to excite the sample and collect the signal from the desired point of the sample. A 515-nm-laser and a 100 × microscope objective with NA = 0.82 were used. The same objective was used to collect the luminescence. The circular polarisation of the PL was analysed using a quarter-wave plate (that maps circular polarisations of opposite helicity into opposite linear polarisations) and a Wollaston prism steering the components of opposite linear polarisation (and thus of opposite helicity) to different lines of the liquid-nitrogen-cooled Si-CCD we employed (100BRX by Princeton Instruments). In this manner, the $\sigma^+$ and $\sigma^-$ components could be measured simultaneously. A monochromator with 0.75 m focal length (Princeton Instruments) and a 600 grooves/mm grating was used to disperse the PL signal.

## Second harmonic generation measurements

The measurements were performed by using the 900 nm line of a tunable pulsed Ti:Sapphire laser with a pulse width of <100 fs and a repetition rate of 80 MHz. The sample was excited and measured under a 50 × confocal objective lens (NA = 0.85), and the frequency-duplicated light was collected in a backscattering configuration. Linear polarisers were used to select laser light and SHG signal with given polarisation states. The sample was placed on a rotation stage to collect the SH response in terms of relative polarisation angles.

## Data availability

The data that support the findings of this study are available from the corresponding authors upon request.

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

## Acknowledgements

The authors thank Andrey Chaves, Paulo E. Faria Jr and Tomasz Woźniak for useful discussions about the magneto-photoluminescence results and are grateful to Daniele Sanvitto for support with the SHG measurements. E.B., A.P., A.B., and M.R.M. acknowledge support by the European Union's Horizon 2020 research and innovation programme through the ISABEL project (No. 871106). This project was funded within the QuantERA II Programme that has received funding from the European Union's Horizon 2020 research and innovation programme under Grant Agreement No. 101017733, and with funding organisations Ministero dell'Università e della Ricerca (A.P. and M.F.) and by Consiglio Nazionale delle Ricerche (G.P.). A.P. and M.F. acknowledge financial support from the PNRR MUR project PE0000023-NQSTI. M.F. and G.P. acknowledge funding from the PRIN2022 project DELIGHT2D (Prot. 20222HNMYE). E.B. acknowledges support from La Sapienza through the grants Avvio alla Ricerca 2021 (Grant no. AR12117A8A090764) and Avvio alla Ricerca 2022 (Grant no. AR2221816B672C03). The authors acknowledge support from the National Science Centre, Poland, through Grants No. 2018/31/B/ST3/02111 (K.O. P. and M.R.M.) and No. 2017/27/B/ST3/00205 (A.B.). K.W. and T.T. acknowledge support from the JSPS KAKENHI (Grant Numbers 19H05790 and 20H00354).

## Author contributions

E.B. and A.P. conceived and supervised the research. E.B. and M.C. fabricated the heterostructures. E.B., F.T., S.C., M.C. and G.C. performed the optical measurements and analysed the data. E.B., A.P., K.O.P., L.K., and M.R.M. performed the magneto-optical measurements, with the support of A.B., and E.B. analysed the data. A.M. provided support for the SHG measurements. M.F. provided support for the time-resolved measurements. G.P. contributed to the sample characterisation. T.T. and K.W. grew the hBN samples. E.B. and A.P. wrote the manuscript. The results and the manuscript were approved by all the coauthors.

## Competing interests

The authors declare no competing interests.
