## [Peer Review File · Nature Communications]

Reviewers' Comments:

Reviewer #1:

Remarks to the Author:

The manuscript by Blundo et al., studied the moire exciton properties, particularly the exciton dynamics upon heating. They found that moire excitons can turn into free interlayer excitons and become truly delocalized excitons at elevated temperatures. These results are interesting and addressed an important question in this field. Overall, this is a high-standard paper containing a lot of high-quality content and in-depth discussions. However, some concerns need to be addressed before the manuscript goes to the next stage. First of all, the underlying reason why moire excitons can de-trap at higher T (> 120 K)? Which is much higher than that in the literature. This is the key observations in this manuscript. The authors, however, provide insufficient explanations and ground evidences to support this key finding. Whether the sample quality, including interlayer twist angle, or defects, have the impacts? In addition, the reversal of the interlayer exciton g -factor is not well explained.

Reviewer #2:

Remarks to the Author:

In this paper, Elena et al. discuss a critical topic in the field of moire physics, which is "Up to which extent moire excitons can be regarded as truly moire-confined?" This topic is of fundamental importance in all the intriguing physics related with moire excitons in TMD heterostructures. In the R-type WSe₂/MoSe₂ heterostructure, the authors firstly repeated the previously reported moire exciton and then unveiled the existence of ladder-like discrete states owing to moire potential through dynamics investigation. By extracting the power index α and the g factor of excitons, a transition from moire exciton to interlayer exciton was confirmed at $T \geq 120$ K. The following dynamics investigation further demonstrate the delocalization of moire excitons at higher temperature.

In general, I find this work reaches a sound and clear conclusion, which identifies the transition from moire excitons to interlayer excitons with multiple approaches. I think this work is certainly of interest to the community and of potential high impact. However, I found a few issues that need to be addressed before the paper can be accepted for publication in Nature Communications. My concerns are described below:

1. The WSe₂/MoSe₂ heterostructure fabricated in this work is covered with top hBN. In case the moire potential is possible to be perturbed by the hBN, can the authors provide some further comments on the relationship between the moire exciton-interlayer exciton transition and hBN encapsulation (no hBN/top hBN only/both top and bottom hBN)? I believe this information will be critical to the community.
2. In Fig. 1c, the authors presented the moire exciton with multiple peaks, with an energy spacing of 12.9 ± 1.1 meV, which agrees quite well with previous research. However, these multiple peaks are seldom discussed in the following sections. I believe the paper will be more informative if the authors provide more information about the power index α and g factor about these moire multiple peaks.
3. Considering the importance of twist angle in moire related physics, the authors only provide a rough estimation of the moire period through the ladder of moire exciton energy. However, this evaluation is based on many assumptions, which leaves the ground of the whole work unsolid. It will be much better if the authors can characterize the crystal axes of WSe₂ and MoSe₂ directly by experimental techniques such as SHG and TEM.

Reviewer #3:

Remarks to the Author:

Measuring the g factor in this system at high temperatures can be highly intriguing, as it could reveal more information about the system. However, it is recommended to conduct the measurements in several higher-quality devices to ensure that the observed effect is intrinsic. Additionally, providing a well-defined explanation for each step of the experiment would help make

the claim more convincing. As of now, the reviewer cannot recommend publishing the paper in its current form.

- The band centred at ≈ 1.36 eV, labelled MX, is due to MX recombination (with the electron and hole being confined in the MoSe₂ and WSe₂ layer, respectively)

In this study, it was found that the energy of the exciton observed is higher than that reported for R-type heterostructures. This elevated energy may be attributed to the absence of a bottom hexagonal boron nitride (hBN) layer, the presence of unintentional doping in the system, or a lower material quality, which can lead to a deviation from the expected excitonic behavior.

- Fig. 1c displays the MX spectrum recorded at $T=6$ K with $P_{exc}=5$ nW (corresponding to 0.64 W/cm²). The spectrum can be deconvoluted into several Gaussian components. The latter are equally spaced by (12.9 ± 1.1) meV, reflecting the quantised states of the moiré potential. The lack of information on the homogeneity of the sample renders the observed phenomenon in this case less convincing in terms of its origin being attributed to the varying energy levels of the moiré potential and the subsequent estimation of the moiré lattice constant.

- The very narrow lines superimposed on the multi-gaussian lineshape of the MX band likely correspond to single MXs confined in just one moiré minimum

It's worth noting that recent research indicates that the narrow lines are likely caused by external factors, such as defects within the system. Since the device is in direct contact with a rough SiO₂ substrate, it's possible that the observed narrow lines have an extrinsic origin

- The excellent alignment leads to a sizeable signal of the HS IXs up to room temperature, as shown in Fig. 1d. Note that the recombination from the HS is indicated as MX ...

The MoSe₂ signal is significant and may indicate either a weak interaction between two monolayers or a large twist angle.

- recombination is centred at a higher energy —by about 40 meV [5, 10, 24–27]— due to the shallower moiré potential for H-type with respect to R-type HSs

Reference <https://www.science.org/doi/10.1126/sciadv.1701696>

- shows the μ -PL spectrum of the investigated WSe₂/MoSe₂ HS recorded at a power 200 times larger ($P_{exc}=1$ μ W, i.e., 128 W/cm²) than in Fig. 1c. This results in a non negligible contribution from a component centred at about 1.4 eV

The power (1 μ W) employed in this section is insufficient to de-trap IX. Furthermore, if the author believes this to be the case, it would be valuable to measure the PL as a function of magnetic field in the system using this power level and demonstrate the change in the g factor at 6K.

- MD MLs the intralayer exciton X is known to have much shorter recombination decay times, on the order of a few ps to a few of tens of ps [28, 29], in contrast with MX. It is worth noting that the spectral range centred at 1.4 eV should be considered as a mixing of the highest energy level of the moiré potential and of the free IX component.

It is recommended to include the laser pulse in the lifetime data. Additionally, the extended lifetime observed at higher power levels may be attributed to the activation of non-radiative channels, as noted in several publications. The origin of the peak at 1.4 is insufficiently discussed, and there is no record of the power or excitation wavelength used in this measurement.

Conducting power-dependent lifetime measurements could clarify the origin further. Moreover, since the data is non-confocal, it may be possible that the exciton is originating from various locations within the sample due to high diffusion in the system. Furthermore, the low quality of the device, as evidenced by the absence of hbn on the bottom and the visible air bubbles in Figure 1a, could also be contributing factors.

- The different states of the moiré potential also present a different formation dynamics. Fig. 2c shows the time evolution of the MX signal up to 1 ns after the laser pulse excitation. In this case, the data are reproduced

It would be preferable to include the laser pulse in the reported data. As the detection is performed throughout the sample, the observed long lifetime could potentially be attributed to the time it takes for the detected exciton to flow within the sample, reach a defect or the edge of the sample, and recombine. Moreover, the influence of IX-IX repulsion has not been taken into account in this case, which could be a source of the observed phenomena. Conducting power-dependent lifetime measurements with a confocal setup could help clarify this ambiguity. As of now, the conclusion drawn from the data seems questionable.

- to $6.25 \cdot 10^{10}$ cm⁻² and a sizeable exciton-exciton interaction is possible thus explaining the decrease in the emission decay time observed in Fig. 2b as well as the MX band blueshift with P_{exc}

The assumption is that when the sample is excited, all the IX traps in the vicinity are also excited. One can view this as exciting the sample at point A, where some IX traps flow out of the light cone due to IX-IX interaction. The IX traps that flow away will eventually become trapped in different sites or defects throughout the sample. Therefore, the density per trap in this case should encompass the entire sample area rather than just the excitation area.

- photoluminescence (PL) spectra of the studied WSe₂/MoSe₂ heterostructure (HS)

The acronyms have already been defined previously, so there is no need to define them again.

- PL integrated intensity dependence on the laser power P_{exc} for MX (azure symbols) and X (dark yellow symbols) bands at $T=6$ K (full symbols) and $T=90$ K (open symbols). Solid and dashed lines are fits to the data with Eq. 3 for $T=6$ K and 90 K, respectively. At $T=6$ K, the α coefficient values are 0.55 ± 0.02 and 0.89 ± 0.02 for MX and X, respectively. At $T=90$ K, the α coefficient values are 0.99 ± 0.02 and 0.97 ± 0.03 for MX and X, respectively

Is the analysis method solely based on integrating the data, or does it involve fitting Lorentzian or Gaussian functions? It would be preferable to provide a more detailed explanation of the analysis method used to examine the data.

- Temperature variation of the α coefficient for the MX-IX and X bands. In the former case, a clear transition from a sublinear to a linear behaviour is found and ascribed to the transition from a moiré localisation regime to a free interlayer exciton one (hence the mixed label MX-IX)

It would be advisable to provide a more comprehensive and detailed explanation to support this claim. Furthermore, it would be beneficial to clarify if the power range for all the measurements is the same. If not, how do you ensure that the observed phenomena are not merely the low-power behavior of the system, which has not been recorded in higher temperatures.

- $T=90$ K μ -PL spectra for different laser excitation powers in the energy region, where the MX and IX recombinations can be simultaneously observed. IX takes over MX upon increase of the photogenerated carrier density. e Same as d for $T=296$ K, where only the IX transition is observable.

If the power is increased to the point where the trapped excitons become free, they should be able to gain momentum and move out of the light cone. In such a scenario, it raises the question of why we still observe them if they are completely free. Moreover, what prevents the free IX-IX from interacting with each other, leading to the disappearance of the signal?

- shown in Supporting Note 5. From $T=6$ K to $T=120$ K, the HS signal is dominated by the MX band, which undergoes a redistribution of the carrier population between the different states of the moiré potential. Starting from $T=120$ K, a high-energy component due to IXs appears and becomes increasingly important relative to the MX band, un

As the temperature is raised, energy is added to the system, causing the trapped excitons to move to higher energy levels. In such a scenario, it is puzzling that the exciton with lower energy level starts to exhibit higher intensity than the one with higher energy level. Additionally, the 80 meV energy difference between the MX and IX is not entirely trustworthy since the sample's inhomogeneity has not been measured or addressed.

- the $\sigma +$ red component is at higher energy than the $\sigma -$ blue one). Then, magneto- μ -PL measurements were performed also at $T = 160$ K ($P_{exc}=75 \mu\text{W}$), where the HS exciton band is instead dominated by free IXs. Fig. 4d shows the $\sigma +$ and $\sigma -$ components of the IX spectra at different magnetic fields.

At 160 K, there is a coexistence of both the well-defined MX and IX. To demonstrate that these two excitons have different origins that are independent of temperature and power, it is recommended to measure the g factor for both at the same temperature. Additionally, displaying the map of IX energy shift by the magnetic field, as seen in Figure 4b for the 160 K data, would be beneficial. Moreover, if the IX is playing a role in the low-temperature PL at high enough power, as claimed, then such a g factor should be observable.

- At $T=100$ K, τ approaches the temporal resolution limit. The higher temperatures and the ensuing MX ionisation process result indeed in a decreased contribution of the moiré localisation step and thus in a reduction of the time required to build up the exciton population contributing to the MX/IX band----

Can it simply be explained through faster flow of IX?

In the method section the wavelength of the excitation for measuring the magnetic field PL is not clarified.

Point-by-point reply to the Referees

Localisation-to-delocalisation transition of moiré excitons in WSe₂/MoSe₂ heterostructures

Elena Blundo,^{1,*} Federico Tuzi,¹ Salvatore Cianci,¹ Marzia Cuccu,¹ Katarzyna
Olkowska-Pucko,² Łucja Kipcza,² Giorgio Contestabile,¹ Antonio Miriametro,¹ Marco
Felici,¹ Giorgio Pettinari,³ Takashi Taniguchi,⁴ Kenji Watanabe,⁵ Adam Babiński,²
Maciej R. Molas,² and Antonio Polimeni^{1,*}

¹ *Physics Department, Sapienza University of Rome, 00185, Roma, Italy.*

² *Institute of Experimental Physics, Faculty of Physics, University of Warsaw, Pasteura 5,
02-093 Warsaw, Poland*

³ *Institute for Photonics and Nanotechnologies (CNR-IFN), National Research Council, 00133,
Rome, Italy*

⁴ *International Center for Materials Nanoarchitectonics, National Institute for Materials
Science, 1-1 Namiki, Tsukuba 305-0044, Japan.*

⁵ *Research Center for Functional Materials, National Institute for Materials Science, 1-1
Namiki, Tsukuba 305-0044, Japan*

* Corresponding authors:

elena.blundo@uniroma1.it,

antonio.polimeni@uniroma1.it

Contents

Reply to Referee 1	1
1.1	1
Reply to Referee 2	7
2.1	7
2.2	9
2.3	12
Reply to Referee 3	13
3.1	13
3.2	15
3.3	16
3.4	19
3.5	20
3.6	20
3.7	21
3.8	23
3.9	24
3.10	24
3.11	24
3.12	25
3.13	27
3.14	27

3.15	28
3.16	30
3.17	31
References	32

Referee 1

The manuscript by Blundo et al., studied the moiré exciton properties, particularly the exciton dynamics upon heating. They found that moiré excitons can turn into free interlayer excitons and become truly delocalized excitons at elevated temperatures. These results are interesting and addressed an important question in this field. Overall, this is a high-standard paper containing a lot of high-quality content and in-depth discussions. However, some concerns need to be addressed before the manuscript goes to the next stage.

We thank the Referee for her/his positive feedback on our manuscript and for highlighting that it addresses a hot topic in the field. We thank the Referee also for her/his specific comments, which we address below.

The newly added or modified parts of our manuscript were highlighted in red for ease of the Referee.

1.1

First of all, the underlying reason why moire excitons can de-trap at higher T (> 120 K)? Which is much higher than that in the literature. This is the key observations in this manuscript. The authors, however, provide insufficient explanations and ground evidences to support this key finding. Whether the sample quality, including interlayer twist angle, or defects, have the impacts? In addition, the reversal of the interlayer exciton g -factor is not well explained.

On general grounds, excitons trapped (or localised) in a potential well would start being de-trapped at temperatures corresponding to a thermal energy of the order of the potential well depth. Although this could be regarded as a simplistic picture, it provides a first estimation of the temperature at which one should expect that localised excitons turn free. As a matter of fact, at $T=120$ K, the thermal energy is about 10 meV that can be sufficiently high to ignite moiré exciton de-trapping in the case of WSe₂/MoSe₂ heterostructures (HSs). In these HSs, the potential depth is about 100 meV and the energy levels of the moiré excitons are few/several tens of meV below the potential edge [1, 2]. Of course, as implied by the Referee, the presence of non-radiative centres energetically close to the moiré potential may reduce considerably the maximum temperature at which moiré (and free interlayer) excitons contribute to the emission spectra. This mechanism is certainly present in our (as well as in any other) samples; indeed the overall emission efficiency of the sample inevitably decreases with increasing temperature. Nevertheless, thanks to the presumably small density of non-radiative centres in our samples, we can pinpoint spectroscopically the transition from a localised to a de-localised carrier regime as T is increased up to room temperature.

Our work apart, another instance can be found in the literature, where signatures of moiré exciton de-trapping at high temperature were observed. In Ref. [2], an exciton diffusivity experiment showed that for $T>100$ K moiré excitons start de-trapping. However, the moiré exciton de-trapping process was not addressed spectroscopically as in our work, nor the transformation into a free interlayer exciton was reported in Ref. [2]. Interestingly, the authors of Ref. [2] investigated the effect of the interlayer twist angle, θ , on the diffusion of the moiré excitons out of their trapping potential. They observed that the larger θ (namely, the lower the moiré potential depth), the higher the exciton de-trapping probability with increasing temperature. This fact points to the important role played by θ , as indeed mentioned by the Referee.

Following the suggestion of the Referee, as well as of the other two Referees, we fabricated another WSe₂/MoSe₂ HS (hereafter, we name this sample HS2), which is doubly encapsulated in h-BN (namely the HS was fabricated on top of a h-BN flake and finally capped by a few nm-thick h-BN flake). Second harmonic generation (SHG) measurements were exploited to fabricate the new HS with a virtually null twist angle. The purpose was to address the role of

the encapsulation on the quality of the HS and thus on the possibility to observe the localised to de-localised exciton transition that takes place at relatively high T values. The HS presented in the first version of the manuscript was fabricated directly on SiO_2 (hereafter, we name this sample HS1) and it was characterised before and after h-BN capping. As a general, we did not observe major variations in the sample quality related to the presence of h-BN (bottom/top) layers nor in the possibility to observe the localised to de-localised exciton transition. On the other hand, HS2 (*i.e.*, the newly fabricated HS) resulted to have a slightly larger θ ($\approx 0.74^\circ$) as compared to HS1 (*i.e.*, the previous HS), where $\theta \approx 0.46^\circ$. First, this was derived by the larger spacing of the moiré energy manifold in HS2 as compared to HS1 (20.3 meV for HS2 *vs* 12.8 meV for HS1), see Fig. 1.1.1a. Following the procedure discussed in Supporting Note 2, such a spacing corresponds to $\approx 0.74^\circ$. Second, HS2 exhibits exciton PL decay times longer than those observed in HS1 (see Fig. 1.1.1b) as expected for larger twist angles, a feature related to the larger k-space mismatch of HS2 with respect to HS1, as nicely discussed in Ref. [3]. Our results agree very well with those reported in Ref. [3] for a $\text{MoSe}_2/\text{WSe}_2$ HS with a 1° twist angle, for which a 21 meV spacing was found between the energy levels. Following the Referee’s remark, the data shown in Fig. 1.1.1 were included and discussed in Supporting Note 2.

Figure 1.1.1: PL properties of HS2. **a** μ -PL spectrum of HS2 acquired at 6 K with very low laser power excitation (10 nW). The spectrum can be reproduced by five Gaussian functions that are spaced by (20.3 ± 3.4) meV. **b** Time-evolution of the μ -PL signal of HS2 recorded in the $\Delta t=0$ -800 ns interval from the laser pulse and comparison with the analogous data acquired for HS1. The excitation power was $1 \mu\text{W}$.

We thus performed a careful laser power-dependent PL study of HS2 for different temperatures (similar to Fig. 3 of the main manuscript for HS1). Indeed, the same phenomenology observed for HS1 was found also for HS2. As noted in Fig. 3d of the main text for HS1, at intermediate temperatures of about 90 K a dramatic lineshape change can be observed when increasing power, showing a clear transition from the MX prevailing at low P_{exc} and the IX prevailing at high P_{exc} . The same trend was indeed observed also for HS2, as demonstrated by the power-dependent spectra acquired at 80 K and shown in Fig. 1.1.2.

Power-dependent measurements on HS2 were performed at different temperatures between 6 K and 296 K, analogously to HS1. The integrated PL intensity of the MX/IX and of the X band were then analysed and fitted by Eq. 3 of the main text, leading to the α coefficients shown in Fig. 1.1.3. Indeed, we found that the moiré exciton de-trapping process takes place at slightly lower T in HS2 as compared to HS1 because of the shallower moiré potential of the former. This is highlighted in Fig. 1.1.3 by the black dashed lines. In the plot concerning HS1, we also added some points which were measured before capping the sample with hBN (*i.e.*, on the bare HS on SiO_2). Indeed, these points follow the same trend of the data after hBN-capping, showing that

Figure 1.1.2: Left: $T=90$ K μ -PL spectra of HS1 for different laser excitation powers in the energy region where the MX and IX recombinations can be simultaneously observed. IX takes over MX upon increase of the photo-generated carrier density. Right: Same at $T=80$ K for HS2.

this process has not an impact on the α vs T behaviour.

Figure 1.1.3: Temperature variation of the α coefficient for the MX-IX and X bands, comparison between HS1 and HS2. For HS1, some points measured before hBN-capping were also included in the plot, showing that the data agree well with those acquired after hBN-capping. Both for HS1 and HS2, a clear transition from a sublinear to a linear behaviour is found for the MX-IS band, which is ascribed to the transition from a moiré localisation regime to a free interlayer exciton one. Such a transition occurs at ≈ 120 K for HS1, and for ≈ 100 K for HS2, as highlighted by the black dashed lines.

These observations address the point raised by the Referee and are reported in the new version of the manuscript. In particular, Figs. 1.1.2 and 1.1.3, along with the whole set of plots showing the dependence of the integrated PL intensity of the MX/IX and X bands on power, from which the α coefficients were derived, were included and discussed in Supporting Note 7. Furthermore, the following text was added in the main manuscript:

A similarly comprehensive study was performed also for HS2, which has a larger twist angle of 0.74° . The results are shown in Supporting Note 7. As for HS1, the X band shows a linear

behavior ($\alpha=1$) all over the temperature range. Although also in HS2 the coefficient α for the MX-IX band exhibits a progressive increase from 0.5 to 1 (see Supporting Note 7), the plateau-value of 1 is reached at a temperature of about 100 K (lower than the 120 K value observed in HS1; see Fig. 3c) consistently with the shallower moiré potential at greater twist angles [4]. On the one hand, these results strengthen the picture described so far. On the other hand, they point to the crucial role of the twist angle in determining the thermal stability of moiré excitons and associated phenomena in moiré superlattices.

Regarding the reversal of the sign of the exciton gyromagnetic factor, g_{exc} , we performed additional/new magneto- μ -PL measurements on HS1 and HS2 employing a magnetic field B up to 16 T (namely, 4 T larger than in the first set of experiments) under different experimental conditions.

The measurements were meant to identify the physical conditions under which the sign reversal of g_{exc} occurs. To this end, we varied the sample temperature and the density of photo-generated carriers $n_{\text{e-h}}$. Indeed, T rules the moiré exciton de-trapping, as we already discussed in the previous version of the manuscript. As reported very recently in a theoretical work —Ref. [5]— which was published after the submission of our manuscript, the increase of $n_{\text{e-h}}$ leads to an augmented exciton-exciton interaction resulting in a decrease of the effective moiré potential that eventually causes an enhancement of the exciton intercell hopping and hence of the exciton delocalisation [5]. Fig. 1.1.4 shows circular polarisation-dependent (σ^\pm) μ -PL spectra recorded at $T=10$ K and $B=16$ T for laser power P_{exc} varying over five orders of magnitude. With in-

Figure 1.1.4: Polarisation-resolved μ -PL power studies performed at 10 K under a 16 T magnetic field for both HS1 and HS2. At low powers, several Zeeman-split narrow lines can be observed, such as those highlighted by the black circle. At high powers, a change in the sign of the Zeeman splitting can clearly be noticed.

creasing P_{exc} (or equivalently $n_{\text{e-h}}$), the positive splitting of the energy (*i.e.*, $E^{\sigma^+} > E^{\sigma^-}$, hence g_{exc} positive) of the narrow emission lines related to various moiré-confined excitons (see, *e.g.*, the narrow lines highlighted by the small circle at the lowest P_{exc} used) becomes nearly zero as the PL band broadens and eventually turns negative on the high energy side of the μ -PL spectrum at the highest P_{exc} used. This result was already presented in Supporting Note 7 of the previous version of the manuscript (the figure has now been moved in the main text as Figure 5), but therein only two P_{exc} values were considered. In light of Ref. [5], the high density

of excitons achieved for large P_{exc} 's leads to an increased intercell exciton hopping, and to an ensuing sizable contribution from de-trapped moiré excitons to the emission spectrum. The sign reversal of the Zeeman splitting can be even better observed for $T > 100$ K, where the free interlayer excitons dominate the emission spectrum, as already reported in the previous version of the manuscript and shown in Fig. 4 of the main text.

Therefore, we have identified the experimental conditions for which g_{exc} changes sign: Moiré-confined excitons exhibit a positive Zeeman splitting and moiré-de-trapped excitons (or free interlayer excitons) feature a negative Zeeman splitting. The reversal of g_{exc} is thus ascribable to this change of character of the exciton emission and corresponding experimental conditions.

In a previous theoretical work [6], g_{exc} was evaluated for atomically reconstructed (*i.e.*, real) WSe₂/MoSe₂ HSs and it was found that the moiré-confined excitons that dominate the emission spectra feature a positive $g_{\text{exc}}=+6.2$ in accordance with our results. On the other hand, de-trapped moiré excitons, or free interlayer excitons, can be regarded as freely moving quasi-particles, whose g_{exc} can be evaluated by summing up the contribution to the gyromagnetic factor coming from the holes (namely, the valence band of WSe₂) and the electrons (namely, the conduction band of MoSe₂) composing the excitons. By taking these theoretically calculated contributions from Ref. [6], one finds $g_{\text{exc}} = -4.44$, as we reported in the previous version of the manuscript.

In summary, the reversal of g_{exc} is deeply related to the nature of the excitonic transitions that dominate the emission spectra for specific experimental conditions. High temperature ($T > 100$ K) and/or elevated photogenerated carrier density ($n_{\text{e-h}} \gtrsim 10^{12} \text{ cm}^{-2}$) favour the spectral predominance of free interlayer excitons with *negative* g_{exc} , while low temperature ($T < 100$ K) and small photogenerated carrier density ($n_{\text{e-h}} < 10^{12} \text{ cm}^{-2}$) conditions make moiré-confined excitons with *positive* g_{exc} to dominate the emission spectra.

Fig. 1.1.4 was added to the revised version of our work as **Supporting Note 12**, and the data which were previously shown in Supporting Note 7 of the first version of the manuscript were moved to the main text of the revised manuscript as Figure 5 (see 1.1.5 of this reply). Overall, the following parts (in red) were added to the manuscript:

Fig. 5a shows a series of μ -PL spectra recorded on HS1 with opposite circular polarisation (σ^+ and σ^-) at different magnetic fields for $T=6$ K and low laser excitation power $P_{\text{exc}}=0.2 \mu\text{W}$. Several narrow lines due to moiré confined excitons can be observed, superimposed on a continuum background. Like in Fig. 4b, the MX lines exhibit a positive Zeeman splitting with average $g_{\text{exc,MX}} = +6.78 \pm 0.11$ (panel c of the figure). Fig. 5b shows the same study of panel a recorded on the same point of the HS but with a P_{exc} value increased by about a factor of 400. The narrow lines associated to moiré confined excitons merge forming a single band, whose maximum shows a blue-shift of more than 20 meV. As discussed in Fig. 3, for high laser power the emission band comprises a mixture of (interacting) moiré confined excitons and free interlayer excitons IXs. Under these circumstances, the ZS value reverses its sign and g_{exc} becomes negative, as shown in Fig. 5c. We also performed a PL study at fixed $B=16$ T by varying P_{exc} over more than three orders of magnitude. The data are described in Supporting Note 12 for both HS1 and HS2 and show how increasing the density of excitons the initially positive ZS of the single moiré exciton progressively reduces and eventually goes negative as the MX component broadens and the IX band on the high energy side of the PL spectrum starts prevailing. These findings provide further evidence of the effective screening of the moiré potential caused by photo-generated carriers [7, 5].

Figure 1.1.5: **a-b** Helicity-resolved normalised μ -PL spectra of HS1 under magnetic field at $T = 6$ K for two different laser excitation powers P_{exc} (focused via a $100\times$ objective with $\text{NA} = 0.75$). The two sets of data were acquired in the same point of the HS. For $P_{\text{exc}} = 0.2 \mu\text{W}$ (**a**), many narrow lines can be seen. M1, M2 and M3 indicate three such narrow lines. At high powers $P_{\text{exc}} = 75 \mu\text{W}$ (**b**) a continuous band can be seen. **c** ZS of the lines M1-M3 of panel **a** and of the MX/IX band of panel **b**, showing an opposite sign of the g -factor. The data of lines M2 and M3 are up-shifted by 1 and 2 meV, respectively (as indicated by the double-sided arrows on the left) for sake of clarity.

We thank the Referee once more for her/his comments, which allowed us to improve our work.

Referee 2

In this paper, Elena et al. discuss a critical topic in the field of moiré physics, which is “Up to which extent moire excitons can be regarded as truly moire-confined?” This topic is of fundamental importance in all the intriguing physics related with moiré excitons in TMD heterostructures. In the R-type WSe₂/MoSe₂ heterostructure, the authors firstly repeated the previously reported moire exciton and then unveiled the existence of ladder-like discrete states owing to moiré potential through dynamics investigation. By extracting the power index α and the g-factor of excitons, a transition from moiré exciton to interlayer exciton was confirmed at $T \gtrsim 120$ K. The following dynamics investigation further demonstrate the delocalization of moiré excitons at higher temperature. In general, I find this work reaches a sound and clear conclusion, which identifies the transition from moiré excitons to interlayer excitons with multiple approaches. I think this work is certainly of interest to the community and of potential high impact. However, I found a few issues that need to be addressed before the paper can be accepted for publication in Nature Communications. My concerns are described below:

We thank the Referee for her/his positive feedback on our manuscript and for highlighting that it address a critical topic in the field of moiré physics. We thank the Referee also for her/his concerns, which we address below.

The newly added or modified parts of our manuscript were highlighted in red for ease of the Referee.

2.1

The WSe₂/MoSe₂ heterostructure fabricated in this work is covered with top hBN. In case the moiré potential is possible to be perturbed by the hBN, can the authors provide some further comments on the relationship between the moiré exciton-interlayer exciton transition and hBN encapsulation (no hBN/top hBN only/both top and bottom hBN)? I believe this information will be critical to the community.

We thank the Referee for this remark.

Following the request of the Referee, as well as those of the other two Referees, we fabricated another WSe₂/MoSe₂ HS (hereafter, we name this sample HS2), which is doubly encapsulated in h-BN (namely the HS was fabricated on top of a h-BN flake and finally capped by a few nm-thick h-BN flake). Second harmonic generation (SHG) measurements were exploited to fabricate the new HS with a virtually null twist angle. The purpose was to address the role of the encapsulation on the quality of the HS and thus on the possibility to observe the localised to de-localised exciton transition that takes place at relatively high T values. The HS presented in the fist version of the manuscript was fabricated directly on SiO₂ (hereafter, we name this sample HS1) and it was characterised before and after h-BN capping. As a general, we did not observe major variations in the sample quality related to the presence of h-BN (bottom/top) layers nor in the possibility to observe the localised to de-localised exciton transition. On the other hand, HS2 (*i.e.*, the newly fabricated HS) resulted to have a slightly larger θ ($\approx 0.74^\circ$) as compared to HS1 (*i.e.*, the previous HS), where $\theta \approx 0.46^\circ$. First, this was derived by the larger spacing of the moiré energy manifold in HS2 as compared to HS1 (20.3 meV for HS2 *vs* 12.8 meV for HS1), see Fig. 2.1.1a. Following the procedure discussed in Supporting Note 2, such a spacing corresponds to $\approx 0.74^\circ$. Second, HS2 exhibits exciton PL decay times longer than those observed in HS1 (see Fig. 2.1.1b) as expected for larger twist angles, a feature related to the larger k-space mismatch of HS2 with respect to HS1, as well discussed in Ref. [3]. Our results agree very well with those reported in Ref. [3] for a MoSe₂/WSe₂ HS with a 1° twist angle, for which a 21 meV spacing was found between the energy levels. Following the Referee’s remark, the data shown in Fig. 2.1.1 were included and discussed in Supporting Note 2.

Figure 2.1.1: PL properties of HS2. **a** μ -PL spectrum of HS2 acquired at 6 K with very low laser power excitation (10 nW). The spectrum can be reproduced by five Gaussian functions that are spaced by (20.3 ± 3.4) meV. **b** Time-evolution of the μ -PL signal of HS2 recorded in the $\Delta t=0$ -800 ns interval from the laser pulse and comparison with the analogous data acquired for HS1. The excitation power was $1 \mu\text{W}$.

We thus performed a careful laser power-dependent PL study of HS2 for different temperatures (similar to Fig. 3 of the main manuscript). Indeed, the same phenomenology observed for HS1 was found also for HS2. As noted in Fig. 3d of the main text for HS1, at intermediate temperatures or about 90 K a dramatic lineshape change can be observed when increasing power, showing a clear transition from the MX prevailing at low P_{exc} and the IX prevailing at high P_{exc} . The same trend was indeed observed also for HS2, as demonstrated by the power-dependent spectra acquired at 80 K and shown in Fig. 2.1.2.

Figure 2.1.2: Left: $T=90$ K μ -PL spectra of HS1 for different laser excitation powers in the energy region where the MX and IX recombinations can be simultaneously observed. IX takes over MX upon increase of the photo-generated carrier density. Right: Same at $T=80$ K for HS2.

Power-dependent measurements on HS2 were performed at different temperatures between 6 K and 296 K, analogously to HS1. The integrated PL intensity of the MX/IX and of the X band were then analysed and fitted by Eq. 3 of the main text, leading to the α coefficients shown in Fig. 2.1.3. Indeed, we found that the moiré exciton de-trapping process takes place at slightly lower T in HS2 as compared to HS1 because of the shallower moiré potential of the former. This is highlighted in Fig. 2.1.3. In the plot concerning HS1, we also added some points which were measured before capping the sample with hBN (*i.e.*, on the bare HS on SiO_2). Indeed, these

points follow the same trend of the data after hBN-capping, showing that this process has not an impact on the α vs T behaviour. These observations address the point raised by the Referee

Figure 2.1.3: Temperature variation of the α coefficient for the MX-IX and X bands, comparison between HS1 and HS2. For HS1, some points measured before hBN-capping were also included in the plot, showing that the data agree well with those acquired after hBN-capping. Both for HS1 and HS2, a clear transition from a sublinear to a linear behaviour is found for the MX-IS band, which is ascribed to the transition from a moiré localisation regime to a free interlayer exciton one. Such a transition occurs at ≈ 120 K for HS1, and for ≈ 100 K for HS2, as highlighted by the black dashed lines.

and are reported in the new version of the manuscript. In particular, Figs. 2.1.2 and 2.1.3, along with the whole set of plots showing the dependence of the integrated PL intensity of the MX/IX and X bands on power, from which the α coefficients were derived, were included and discussed in Supporting Note 7. Furthermore, the following text was added in the main manuscript:

A similarly comprehensive study was performed also for HS2, which has a larger twist angle of 0.74° . The results are shown in Supporting Note 7. As for HS1, the X band shows a linear behavior ($\alpha=1$) all over the temperature range. Although also in HS2 the coefficient α for the MX-IX band exhibits a progressive increase from 0.5 to 1 (see Supporting Note 7), the plateau-value of 1 is reached at a temperature of about 100 K (lower than the 120 K value observed in HS1; see Fig. 3c) consistently with the shallower moiré potential at greater twist angles [4]. On the one hand, these results strengthen the picture described so far. On the other hand, they point to the crucial role of the twist angle in determining the thermal stability of moiré excitons and associated phenomena in moiré superlattices.

2.2

In Fig. 1c, the authors presented the moiré exciton with multiple peaks, with an energy spacing of 12.9 ± 1.1 meV, which agrees quite well with previous research. However, these multiple peaks are seldom discussed in the following sections. I believe the paper will be more informative if the authors provide more information about the power index α and g-factor about these moiré multiple peaks.

We thank the Referee for the question. What the Referee proposes would indeed be interesting. As for the power-dependent analysis, however, one has to consider that only in the low-power regime the spectrum is characterised by the dominant presence of moiré excitons. When the excitation power is increased, instead, exciton-exciton interactions start to play a role, and at high enough power free IX starts to be created (see Fig. 3a in the main text) in agreement with recent calculations [5]. This makes it not possible to highlight the moiré multiple peaks and follow their behaviour throughout the power study.

As for the g -factor analysis, our previous sets of ZS data were taken at powers not less than 200 nW. This power is indeed much bigger than that of Fig. 1c in the main text, showing the fitting with 5 gaussians. We thus performed further magneto-PL measurements, using a small power of about 10 nW. From this new set of data, we first measured the g -factor of 5 different individual narrow lines (indicated as Ms), finding values between +6.57 and +6.82, see Fig. 2.2.1. We then tried to fit the whole set of data with multiple gaussians, as exemplified in panel **b** at 0 and 12 T. Indeed, the 5 peaks feature a positive g -factor, similarly to the moiré single lines, but with smaller absolute value. This might possibly be attributed to the role played by exciton-exciton interactions. These results are particularly significant since the similar values (sign and modulus) of the g -factor of the exciton manifold and of the single lines indicate a common electronic structure/origin of those recombination lines. In particular, it indicates that the narrow lines are indeed single moiré excitons giving rise to a gaussian distribution when a sufficiently high number of them is photogenerated. As a matter of fact, the exact nature of those narrow lines is a topic recently debated in the literature [2, 8].

Following the Referee's remark, Fig. 2.2.1 and related discussion were included in **Supporting Note 10**, and the following text was added to the main manuscript:

For HS1, we also derived the ZS of the five gaussians into which the moiré exciton band can be deconvolved at low T and P_{exc} , as shown in Fig. 1c. As discussed in Supporting Note 10, in comparison to the single MX lines of Fig. 4d, a smaller ZS is found for the gaussian components resulting in an average $g_{exc} = +4.43 \pm 0.89$. The smaller g_{exc} for the gaussian components might be caused by exciton-exciton interactions. In fact, the gaussian lineshape could be the consequence not only of a distribution of an ensemble of single moiré lines but also of an exciton interaction-induced broadening of the the moiré emission itself.

Figure 2.2.1: g -factor analysis of the moiré energy levels. **a** Polarisation-resolved magneto- μ -PL spectra acquired with $P_{\text{exc}} = 10$ nW from 0 to 12 T in steps of 0.5 T. Five moiré narrow lines (Ms) are indicated. **b** Exemplifying spectra at 12 T and 0 T fitted by 5 gaussian peaks, for both the σ^- and σ^+ polarisations. **c** Top: ZS of the five gaussian peaks (ps) as a function of the magnetic field. The solid lines are linear fits to the data that provide the g -factors displayed in panel **d**. The intercept was subtracted from the data and they were stacked by 4 meV for ease of comparison. Bottom: Same for the 5 different moiré narrow lines. **d** Summary of the g -factors of both the narrow lines and the gaussian components. The error bars are smaller than (comparable to) the point size for the Ms (ps).

2.3

Considering the importance of twist angle in moiré related physics, the authors only provide a rough estimation of the moiré period through the ladder of moiré exciton energy. However, this evaluation is based on many assumptions, which leaves the ground of the whole work unsolid. It will be much better if the authors can characterize the crystal axes of WSe₂ and MoSe₂ directly by experimental techniques such as SHG and TEM.

We understand the Referee's point and we thus made an effort to include further measurements. We had not previously performed SHG measurements because our laser for SHG is unfortunately broken (and the process to buy a new one is proceeding quite slowly). Following the Referee's remark, we succeeded in borrowing a laser from some colleagues for some weeks and we were thus able to perform such measurements. The results are displayed in Fig. 2.3.1. The

Figure 2.3.1: SHG normalised intensity measured in the WSe₂ and MoSe₂ MLs that constitute the HS as a function of the rotation angle of the sample.

measurements were performed while keeping the polarisation of the detected signal and that of the excitation laser fixed and parallel. The sample was then rotated by an angle θ with respect to an arbitrary set of laboratory coordinates (X,Y). Fig. 2.3.1 displays the SHG measurements taken on pieces of the WSe₂ and MoSe₂ MLs that stick out of the HS. The two sets of data were then fitted by the equation:

$$I_{\text{SHG}} = [A \cdot \cos(3\theta')]^2, \quad \theta' = \theta - \theta_0 \quad (2.1)$$

where θ' is the angle between the excitation laser polarisation and the armchair direction, and θ_0 defines the direction of the armchair lattice direction with respect to the X axis of the laboratory system. Indeed, the θ_0 angles found for WSe₂ and MoSe₂ are very close to each other, with a relative twist $\Delta\theta_0 = 0.1^\circ \pm 1.53^\circ$. This confirms that our HS is characterised by a very small twist angle close to zero.

We included the above data and discussion in Supporting Note 2, and we added the following text (in red) to the main manuscript:

From the spacing between the MX states, as detailed in Supporting Note 2, we estimate a moiré superlattice period a_m of about 40 nm, which corresponds to $\theta = (0.46_{-0.04}^{+0.05})^\circ$ [9]. **Second harmonic generation (SHG) measurements confirm that estimation and provide $\theta = 0.1^\circ \pm 1.5^\circ$, as reported in Supporting Note 2. Given the large uncertainty of the SHG data, we will assign to this HS the twist angle $\theta = 0.46^\circ$ determined by the energy spacing of the moiré potential resonances.**

We thank the Referee once more for her/his comments, which allowed us to improve our work.

Referee 3

Measuring the g -factor in this system at high temperatures can be highly intriguing, as it could reveal more information about the system. However, it is recommended to conduct the measurements in several higher-quality devices to ensure that the observed effect is intrinsic. Additionally, providing a well-defined explanation for each step of the experiment would help make the claim more convincing. As of now, the reviewer cannot recommend publishing the paper in its current form.

We thank the Referee for highlighting that measuring the g -factor at high temperature can reveal novel information on these systems and we thank the Referee for raising many questions and concerns, by addressing which we had the opportunity to improve our work. In general, we understand the Referee's concern regarding the need to measure a higher quality sample to ensure the general validity of our results. Thus, we prepared a new WSe₂/MoSe₂ heterostructure (named HS2), which is encapsulated by h-BN (namely, deposited on an h-BN thick flake and capped by a thin h-BN layer) and featuring a similar (just slightly larger) twist angle. This allowed us to strengthen the general validity of our results and address the dependence of our findings on the twist angle and h-BN encapsulation. Regarding the last issue, in our reply to the Referee, we also provide a comparison of our data before and after h-BN capping on the HS presented in the first version of the manuscript (named HS1). In the following, we address the specific points raised by the Referee.

The newly added or modified parts of our manuscript were highlighted in red for ease of the Referee.

3.1

The band centred at ≈ 1.36 eV, labelled MX, is due to MX recombination (with the electron and hole being confined in the MoSe₂ and WSe₂ layer, respectively). In this study, it was found that the energy of the exciton observed is higher than that reported for R-type heterostructures. This elevated energy may be attributed to the absence of a bottom hexagonal boron nitride (hBN) layer, the presence of unintentional doping in the system, or a lower material quality, which can lead to a deviation from the expected excitonic behavior.

We addressed this remark by performing first an analysis of the data reported in the literature relative to WSe₂/MoSe₂ heterostructures (HSs). Figure 3.1.1(a) shows the photoluminescence (PL) peak energies, E_{PL} , of WSe₂/MoSe₂ HSs with the corresponding reference to the literature. All PL data were recorded at low temperature ($T < 20$ K) and the HSs were encapsulated by h-BN, but papers indicated as g, k and q in the figure legend. We grouped the data into two sets depending on the relative alignment angle: *i*) R-type HSs corresponding to a twist angle, θ , around 0° and *ii*) H-type HSs corresponding to θ around 60°. We note that the E_{PL} values relative to each type of HS remain rather distinct, although a large spread of values is observed within each type of HS. Figure 3.1.1(b) shows the dependence on θ of E_{PL} for R-type HSs (namely, the type investigated in our work). The plot includes E_{PL} values for those works, where θ was given. Clearly, no well-defined trend can be overall observed. This is likely due to the large uncertainties in the determination of θ , local deviations from a regular structure caused by imperfections and local strains accompanied by important atomic level reconstruction that result in a non-ideal/regular moiré potential [10].

As pointed out by the Referee, the R-type HSs considered in our work feature E_{PL} values, which are higher than the average reported values, yet well within the literature distribution of E_{PL} for R-type HSs (see Fig. 3.1.1). Furthermore, as shown in Fig. 3.1.2, no major effects related to different h-BN capping sequences could be noticed in our case. Therefore, we exclude that our relatively high E_{PL} values are due to the absence of bottom h-BN capping, nor we

Figure 3.1.1: **a** Survey of the photoluminescence (PL) peak energies, E_{PL} , of WSe₂/MoSe₂ HSs as reported in the literature. R and H indicate HSs with twist angle $\theta \approx 0^\circ$ and $\theta \approx 60^\circ$, respectively. **b** Same data as in **a** for R-type HSs as a function of θ . The plot shows only data for which the twist angle is provided by the authors. References. a:[8]; b:[9]; c:[11]; d:[12]; e:[2]; f:[13]; g:[14]; h:[7]; i:[15]; j:[1]; k:[16]; l:[17]; m:[18]; n:[3]; o:[19]; p:[20]; q:[21]; r:[22].

have evidences of relevant doping effects (our PL spectrum lineshapes appear very similar to those reported in the current literature; see, *e.g.* Refs. [1, 3]). As far as the sample quality is concerned (we assume that the Referees refers to the HS emission efficiency), it is hard to relate this characteristic to the emission energy and, in any event, we observe sizable PL emission from indirect excitons up to room temperature, thus indicating a low density of non-radiative centres.

Figure 3.1.2: μ -PL spectra acquired at 6 K and with analogous power on HS1 (deposited on a SiO₂ substrate) and on HS2 (deposited on a hBN flake) before and after hBN-capping.

Figs. 3.1.2 and 3.1.1 were included in Supporting Note 1, and the main manuscript was modified by adding the following text in red:

The **two** investigated WSe₂/MoSe₂ HSs were fabricated by first depositing the first ML on the substrate (directly on a SiO₂/Si substrate in the case of HS1, on a h-BN flake deposited on a SiO₂/Si substrate in the case of HS2) and by then depositing the second ML on top. The HSs were then capped with a thin h-BN layer to prevent oxidation; see Methods for other details. A comparison of the PL properties before and after the h-BN capping (see Supporting Note 1) did not show sizeable variations of the HS optical properties. More interestingly, no remarkable differences are found between the optical properties of the two HSs, despite the fact that HS1 is directly in contact with the SiO₂/Si substrate while HS2 lies on a h-BN flake. Both HSs are

characterised by a rather homogeneous PL signal, both in terms of lineshape and intensity, see Supporting Note 1.

3.2

"Fig. 1c displays the MX spectrum recorded at $T = 6$ K with $P_{exc} = 5$ nW (corresponding to 0.64 W/cm²). The spectrum can be deconvoluted into several Gaussian components. The latter are equally spaced by (12.9 ± 1.1) meV, reflecting the quantised states of the moiré potential." The lack of information on the homogeneity of the sample renders the observed phenomenon in this case less convincing in terms of its origin being attributed to the varying energy levels of the moiré potential and the subsequent estimation of the moiré lattice constant.

We thank the Referee for this pertinent remark. Indeed, at an early stage of our experiments, we performed a preliminary study of the sample homogeneity. We performed an analogous study also on the new HS (HS2). Fig. 3.2.1 shows such a study performed on both HS2 (the new HS

Figure 3.2.1: **a-b** Colour plot (a) and normalised stacked spectra (b) of the μ -PL measurements performed while scanning the laser ($P_{exc} = 0.22$ μ W) over HS2 with an S-like scan. The scanning step was of about 0.3 μ m. **c-d** Same for HS1. The drops in the intensity of the MX (and corresponding enhancement of the intralayer excitons signal), which can be noticed especially for HS2, are due to the fact that the HS edge was reached. Apart from such drops, the intensity and lineshape of the MX is pretty uniform.

encapsulated in h-BN) and HS1 (the HS that was presented in the first version of the manuscript and that was fabricated directly on the SiO₂/Si substrate). The measurements were taken by performing an S-scan on the central part of the HSs, with steps of about 0.3 μm . The intensity drops revealed by the colour plots are due to the fact that the scan reached the edge of the HS. Overall, both HSs show a very good degree of homogeneity as far as the resonance energy distances and relative spectral weight are concerned. A bit larger degree of fluctuation is instead observed regarding the emission intensity.

Our study reveals that the sample homogeneity is indeed such that the results discussed in the paper hold a general validity and are not related to some specific spots in the sample. Fig. 3.2.1 was included in Supporting Note 1.

3.3

"The very narrow lines superimposed on the multi-gaussian lineshape of the MX band likely correspond to single MXs confined in just one moiré minimum" It's worth noting that recent research indicates that the narrow lines are likely caused by external factors, such as defects within the system. Since the device is in direct contact with a rough SiO₂ substrate, it's possible that the observed narrow lines have an extrinsic origin

The Referee raises an important point about the origin of the narrow PL lines usually observed in transition metal dichalcogenide HSs. This issue was recently discussed in two interesting papers that considered a similar experimental approach but with seemingly contrasting conclusions. The first paper is by Z. Li *et al.* [2]. The second paper is by F Mahdikhanyarvejahany *et al.* [8]. In addition, a theoretical paper by H. Guo *et al.* [23] addressed the position of defect states relative to the moiré potential depth in MoS₂/WS₂ HSs. The authors of Ref. [2] and [8] compared the PL properties of WSe₂/MoSe₂ and WSe₂/h-BN/MoSe₂ HSs. In the latter case, a thin layer of hexagonal boron nitride (h-BN) was inserted in order to suppress the moiré potential and thus establish whether the observed narrow lines are ascribable to the recombination of moiré-confined excitons or defect states (either intrinsic or extrinsic). The absence of narrow peaks in the h-BN-intercalated HS, as well as the observation of a decreasing number of those peaks with increasing the twist angle (namely, with the progressive decrease of the radiative efficiency of the HS), led the authors of Ref. [2] to conclude that the narrow lines are indeed quantised excitonic states associated to the moiré potential. On the other hand, in Ref. [8] it was reported the observation of a distribution of different narrow lines in both the h-BN-intercalated and h-BN-free HS suggesting that those lines do not originate solely from the moiré potential. As in our case, the authors of Ref. [8] performed magneto-PL measurements and found an opposite sign of the Zeeman splitting of the narrow lines in the two types of HSs. In the h-BN-free HS, the gyromagnetic factor of the narrow lines was positive (a feature common to moiré-confined excitons in R-type HSs; see *e.g.* Refs. [18] and [17]). Instead, a negative gyromagnetic factor was observed in the h-BN-intercalated HS as commonly observed in free excitons in TMD monolayers (see [6]).

We believe that the narrow lines can indeed be ascribed to single moiré excitons based on various evidences.

First, the gyromagnetic factor of all those lines is positive as shown in Fig. 3.3.1, while one should expect a mixed behaviour (*i.e.*, both positive and negative gyromagnetic factor) depending on the exciton nature being moiré-type or interlayer exciton localised by defects. Furthermore, from a careful lineshape analysis of the moiré exciton band it is possible to derive a positive gyromagnetic factor for the gaussian components of the moiré band, as shown in Fig. 3.3.1. This suggests that the exciton manifold and the single lines share a similar gyromagnetic factor and thus a similar electronic structure/origin.

Fig. 3.3.1 and related discussion were included in Supporting Note 10, and the following text was added to the main manuscript:

Figure 3.3.1: g -factor analysis of the moiré energy levels. **a** Polarisation-resolved magneto- μ -PL spectra acquired with $P_{\text{exc}} = 10$ nW from 0 to 12 T in steps of 0.5 T. Five moiré narrow lines (Ms) are indicated. **b** Exemplifying spectra at 12 T and 0 T fitted by 5 gaussian peaks, for both the σ^- and σ^+ polarisations. **c** Top: ZS of the five gaussian peaks (ps) as a function of the magnetic field. The solid lines are linear fits to the data that provide the g -factors displayed in panel **d**. The intercept was subtracted by the data and they were stacked by 4 meV for ease of comparison. Bottom: Same for the 5 different moiré narrow lines. **d** Summary of the g -factors of both the narrow lines and the gaussian components. The error bars are smaller than (comparable to) the point size for the Ms (ps).

For HS1, we also derived the ZS of the five gaussians into which the moiré exciton band can be deconvolved at low T and P_{exc} , as shown in Fig. 1c. As discussed in Supporting Note 10, in comparison to the single MX lines of Fig. 4d, a smaller ZS is found for the gaussian components resulting in an average $g_{\text{exc}} = +4.43 \pm 0.89$. The smaller g_{exc} for the gaussian components might be caused by exciton-exciton interactions. In fact, the gaussian lineshape could be the consequence not only of a distribution of an ensemble of single moiré lines but also of an exciton interaction-induced broadening of the the moiré emission itself.

Second, based on the calculations reported in Ref. [23], the potential associated to a chalcogen atom vacancy (one of the most common defect in TMDs) is deeper than the moiré potential in MoS₂/WS₂ HSs, while in WSe₂/MoSe₂ HSs the moiré potential becomes comparable to that of the defect. Nevertheless, it would be a rather fortuitous coincidence that the moiré and the defect states are resonant. In addition, we could expect that, as in semiconductor quantum wells, defect states should be energetically located *below* that of the intrinsic confined excitons.

Finally, the narrow lines are observed both in HS1 (*no* h-BN between the SiO₂ substrate and the heterostructure) and HS2 (h-BN in between the SiO₂ substrate and the heterostructure) thus excluding the hypothesis of the Referee that the narrow lines are due to the fact that "*the device is in direct contact with a rough SiO₂ substrate*".

This can be noticed, for instance, in Fig. 3.3.2, where we show very low power (5 nW) μ -PL spectra acquired at 6 K with high spectral resolution (we used a monochromator with a focal length of 75 cm) for both HS1 (in direct contact with the SiO₂ substrate) and HS2 (deposited on a h-BN layer). Regardless of the substrate of the HS, the two spectra are qualitatively very similar. As already discussed, the spectra can be fitted with 5 gaussians, whose spacing S differs between the two samples (it is larger for HS2) due to the different twist angle (larger for HS2). The very narrow lines that make up the broader Gaussian peaks correspond to single MXs recombining in moiré minima and can be clearly observed for both HSs.

Figure 3.3.2: μ -PL spectra of the MX band acquired with very low laser power excitation (5 nW). The spectrum can be reproduced by five Gaussian functions that are spaced on average by (12.8 ± 1.3) meV (the individual spacings are 11.4, 13.2, 12.2, 14.3 meV) for HS1, and by (20.3 ± 3.4) meV (the individual spacings are 22.5, 17.1, 23.8, 17.6 meV) for HS2. The very narrow lines that make up the broader Gaussian peaks correspond to single MXs recombining in moiré minima.

Fig. 3.3.2 and related discussion were included in Supporting Note 2.

3.4

"The excellent alignment leads to a sizeable signal of the HS IXs up to room temperature, as shown in Fig. 1d." Note that the recombination from the HS is indicated as MX ... The MoSe₂ signal is significant and may indicate either a weak interaction between two monolayers or a large twist angle.

It's not clear to us what the Referee means by noticing that in Fig. 1d "the recombination from the HS is indicated as MX ...". In fact, in Fig. 1d the recombination from the HS at 6 K is indicated as MX, while at room temperature it is indicated as IX. Indeed, at low T the recombination is mostly due to moiré excitons (with a possible contribution from free IXs at ~ 1.4 eV).

As for the signal of the intralayer exciton band of the MoSe₂ monolayer, its spectral weight is strongly dependent on the laser power employed. As a matter of fact, Figs. 3a and 3b of the manuscript show that only at high powers of about 100 μ W the integrated area of the intralayer excitons becomes comparable to that of the interlayer exciton band, while for $P_{\text{exc}} \lesssim 1$ μ W the MoSe₂ signal is much smaller than that of the HS, as expected by the Referee. This is a clear signature that while free intralayer excitons, Xs, may be created with a virtually infinite density, moiré excitons stem from states with a finite density (*i.e.*, the minima of the moiré potential) and thus their PL intensity reaches a saturation with increasing P_{exc} . We point out that the strong interlayer interaction of our HS is evidenced by the observation of the interlayer exciton recombination up to room temperature.

As for the twist angle, it is indeed provided by the spacing of the energy levels of the moiré. To get further evidence, following the remarks of the Referee and of the second Referee, we performed second harmonic generation (SHG) measurements. The results are displayed in Fig. 3.4.1. The measurements were performed while keeping the polarisation of the detected signal

Figure 3.4.1: SHG normalised intensity measured in the WSe₂ and MoSe₂ MLs that constitute the HS as a function of the rotation angle of the sample.

and that of the excitation laser fixed and parallel. The sample was then rotated by an angle θ with respect to an arbitrary set of laboratory coordinates (X,Y). Fig. 3.4.1 displays the SHG measurements taken on pieces of the WSe₂ and MoSe₂ MLs that stick out of the HS. The two sets of data were then fitted by the equation:

$$I_{\text{SHG}} = [A \cdot \cos(3\theta')]^2, \quad \theta' = \theta - \theta_0 \quad (3.2)$$

where θ' is the angle between the excitation laser polarisation and the armchair direction, and θ_0 defines the direction of the armchair lattice direction with respect to the X axis of the laboratory system. Indeed, the θ_0 angles found for WSe₂ and MoSe₂ are very close to each other, with a relative twist $\Delta\theta_0 = 0.1^\circ \pm 1.53^\circ$. This confirms that our HS is characterised by a very small twist angle close to zero.

We included the above data and discussion in Supporting Note 2, and we added the following text (in red) to the main manuscript:

From the spacing between the MX states, as detailed in Supporting Note 2, we estimate a moiré superlattice period a_m of about 40 nm, which corresponds to $\theta = (0.46_{-0.04}^{+0.05})^\circ$ [9]. Second harmonic generation (SHG) measurements confirm that estimation and provide \$\theta = 0.1^\circ \pm 1.5^\circ\$, as reported in Supporting Note 2. Given the large uncertainty of the SHG data, we will assign to this HS the twist angle \$\theta = 0.46^\circ\$ determined by the energy spacing of the moiré potential resonances.

3.5

"recombination is centred at a higher energy —by about 40 meV [5, 10, 24–27]— due to the shallower moiré potential for H-type with respect to R-type HSs"

Reference <https://www.science.org/doi/10.1126/sciadv.1701696>

We followed the Referee's suggestion and included the pertinent work by Y. Yu and co-authors, who reported calculations of the moiré potential, in this specific part of the manuscript.

3.6

"shows the μ -PL spectrum of the investigated WSe₂/MoSe₂ HS recorded at a power 200 times larger ($P_{exc}=1 \mu W$, i.e., 128 W/cm²) than in Fig. 1c. This results in a non negligible contribution from a component centred at about 1.4 eV" The power (1 μW) employed in this section is insufficient to de-trap IX. Furthermore, if the author believes this to be the case, it would be valuable to measure the PL as a function of magnetic field in the system using this power level and demonstrate the change in the g factor at 6K.

Based on the estimation of the photo-excited carrier density detailed in Supporting Note 4, $P_{exc}=1 \mu W$ would correspond to about $n_{e-h}=10^{12} \text{ cm}^{-2}$. According to the recent work by S. Brem and E. Malic [5], the latter value would be sufficient to delocalise moiré-trapped excitons already at cryogenic temperature. As a matter of fact, in Supporting Note 7 of the previous version of the manuscript we already showed the magnetic field dependence of the MX-IX PL band recorded at $P_{exc}=75 \mu W$ (corresponding to about $n_{e-h}=2 \cdot 10^{13} \text{ cm}^{-2}$, which is above the threshold quoted in Ref. [5] to have significant de-trapping of moiré exciton). Under such excitation conditions, the Zeeman splitting analysis displayed in Supporting Note 7 of the previous version of our work yields a *negative* gyromagnetic factor equal to -6.93 as indeed mentioned by the Referee. Furthermore, we performed additional/new magneto-micro-photoluminescence (μ -PL) measurements on HS1 and HS2 employing a magnetic field B up to 16 T (namely, 4 T larger than in the first set of experiments) with P_{exc} values ranging over five orders of magnitude. The results are shown in Fig. 3.6.1. With increasing P_{exc} (or equivalently n_{e-h}), the positive splitting of the energy (i.e., $E^{\sigma^+} > E^{\sigma^-}$, hence g_{exc} positive) of the narrow emission lines related to various moiré-confined excitons (see, e.g., the narrow lines highlighted by the small circle at the lowest P_{exc} used) becomes nearly zero as the PL band broadens and eventually turns negative on the high energy side of the μ -PL spectrum at the highest P_{exc} used. Thus, the high density of excitons achieved for large P_{exc} 's leads to an increased intercell exciton hopping [5], and to an ensuing sizable contribution from de-trapped moiré excitons to the emission spectrum that reflects in the change of sign of g_{exc} . The sign reversal of the Zeeman splitting can be even better observed for $T > 100$ K, where the free interlayer excitons dominate the emission spectrum, as already reported in the previous version of the manuscript and shown in Fig. 4 of the main text.

In the revised version of the manuscript, we better highlighted this point by moving previous Figure 7.1 of Supporting Note 7 of the first version of the manuscript and related discussion to the main text (see Figure 5 in the revised manuscript). We also added Supporting Note 12 that shows the magneto-PL *vs* P_{exc} data discussed above in Fig. 3.6.1. The following text was added

Figure 3.6.1: Polarisation-resolved μ -PL power studies performed at 10 K under a 16 T magnetic field for both HS1 and HS2. At low powers, several Zeeman-split narrow lines can be observed, such as those highlighted by the black circle. At high powers, a change in the sign of the Zeeman splitting can clearly be noticed.

to the main manuscript:

Fig. 5a shows a series of μ -PL spectra recorded on HS1 with opposite circular polarisation (σ^+ and σ^-) at different magnetic fields for $T=6$ K and low laser excitation power $P_{\text{exc}}=0.2$ μW . Several narrow lines due to moiré confined excitons can be observed, superimposed on a continuum background. Like in Fig. 4b, the MX lines exhibit a positive Zeeman splitting with average $g_{\text{exc,MX}} = +6.78 \pm 0.11$ (panel c of the figure). Fig. 5b shows the same study of panel a recorded on the same point of the HS but with a P_{exc} value increased by about a factor of 400. The narrow lines associated to moiré confined excitons merge forming a single band, whose maximum shows a blue-shift of more than 20 meV. As discussed in Fig. 3, for high laser power the emission band comprises a mixture of (interacting) moiré confined excitons and free interlayer excitons IXs. Under these circumstances, the ZS value reverses its sign and g_{exc} becomes negative, as shown in Fig. 5c. We also performed a PL study at fixed $B=16$ T by varying P_{exc} over more than three orders of magnitude. The data are described in Supporting Note 12 for both HS1 and HS2 and show how increasing the density of excitons the initially positive ZS of the single moiré exciton progressively reduces and eventually goes negative as the MX component broadens and the IX band on the high energy side of the PL spectrum starts prevailing. These findings provide further evidence of the effective screening of the moiré potential caused by photo-generated carriers [7, 5].

3.7

"TMD MLs the intralayer exciton X is known to have much shorter recombination decay times, on the order of a few ps to a few of tens of ps [28, 29], in contrast with MX. It is worth noting that the spectral range centred at 1.4 eV should be considered as a mixing of the highest energy level of the moiré potential and of the free IX component." It is recommended to include the laser pulse in the lifetime data. Additionally, the extended lifetime observed at higher power levels may be attributed to the activation of non-radiative channels, as noted in several publications. The origin of the peak at 1.4 is insufficiently discussed, and there is no record of the power or excitation wavelength used in this measurement. Conducting power-dependent lifetime measurements

could clarify the origin further. Moreover, since the data is non-confocal, it may be possible that the exciton is originating from various locations within the sample due to high diffusion in the system. Furthermore, the low quality of the device, as evidenced by the absence of hbn on the bottom and the visible air bubbles in Figure 1a, could also be contributing factors.

We thank the Referee for this remark. We have included the laser pulse in Fig. 6 of the manuscript. As for the data in Figure 2, we have added a figure in Supporting Note 3 to show a comparison between the data and the laser pulse on a suitable time scale. We respectfully notice that the excitation power related to the data in Figs. 2 and 6 was already reported either in the plots or in the caption, and the laser wavelength was already indicated in the Methods section.

The origin of the component at 1.4 eV shown in Fig. 2a of the manuscript was attributed to a prevalent contribution of free interlayer excitons, IXs. This assignment was made by observing the much reduced decay time of that spectral component in comparison to those lying at lower energy due to moiré excitons, as detailed in the discussion of Fig. 2b of the manuscript. Here, such attribution is further supported by the new measurements performed at $T=10$ K and $B=16$ T as a function of P_{exc} ; see Fig. 3.6.1. As already shown in the previous point and discussed in Supporting Note 12, at high P_{exc} the Zeeman splitting of the component at $\gtrsim 1.4$ eV is negative (*i.e.*, $g_{\text{exc}} < 0$) and opposite to that of the moiré excitons ($g_{\text{exc}} > 0$) observed at lower P_{exc} on the low energy component of the band. This is similar to what found in the magneto-PL measurements shown in Fig. 4 of the manuscript, where at $T \gtrsim 100$ K free IXs prevail and $g_{\text{exc}} < 0$.

The Referee suggests to conduct power-dependent lifetime measurements. Actually, those measurements were presented in Fig. 3.2 of the previous Supporting Note 3 (now Supporting Note 4), although they were not explicitly discussed in the main text. We have now discussed them in more detail by adding the following statement:

Power-dependent tr- μ -PL measurements on both HSs reported in Supporting Note 4 and 5 show a progressive shortening of the time decay of the MX band as P_{exc} increases. This is a likely consequence of exciton-exciton interactions, that tend to diminish the exciton lifetime [24, 25], and of the increasing contribution to the MX band from de-trapped IXs with reduced lifetime, as discussed above.

Indeed, the decay time clearly decreases with increasing P_{exc} suggesting that carrier interaction effects contribute most likely in reducing the exciton lifetime. In the revised version of the manuscript, we referred to those results more explicitly. We would like to clarify that all time-dependent and steady-state measurements were performed employing a confocal set-up as it was already reported both in the Results and Discussion section (in the subsection "Moiré exciton dynamics at low temperature", it was written: "*Cw and tr- μ -PL measurements were carried out at variable laser excitation power P_{exc} and temperature T using a confocal microscope setup*") and in the Methods section ("*A 100 \times long-working-distance Zeiss objective with $NA = 0.75$ was employed to excite and collect the light, in a backscattering configuration using a confocal setup*") of the previous version of the manuscript.

We don't understand what the Referee means when she/he says "*the extended lifetime observed at higher power levels may be attributed to the activation of non-radiative channels, as noted in several publications*". In particular, we are dubious that the Referee meant "reduced" instead of "extended". Indeed, high power leads to carrier-carrier interaction effects acting similar to non-radiative channels that eventually reduce the radiative lifetime, as pointed out above.

We also wish to remark that the assumption of the Referee that our HS1 is a "*low quality device*" is based on morphological aspects, solely. As a matter of fact, the results obtained on the h-BN encapsulated HS2 sample are of as "*high quality*" as those obtained on the HS1 sample. The electronic and optical characteristics of our samples compare favourably with those

reported in the current literature in terms of lineshape and intensity: we detected signals with excitation powers as low as few nW even in the bore of the magnet, where optical alignment is much more critical, and up to room temperature that is less frequently reported.

Following the Referee's remark, we performed power-dependent time-resolved PL experiments also on HS2. In Fig. 3.7.1, we compare the decay times and corresponding weights estimated for HS2 with those obtained for HS1 (and already reported in Supporting Note 4). Indeed, very

Figure 3.7.1: Decay times and weights derived from time-resolved μ -PL measurements performed at 6 K for different excitation powers P_{exc} . The data were acquired in the $\Delta t=0-800$ ns interval from the laser pulse. The detection energy was set at the MX-IX band. **a** Summary of the estimated decay times $\tau_{d,n}$ as a function of P_{exc} (top) and corresponding weights $w_{d,n}$ (bottom), for HS1. **b** Same for HS2.

similar trends are observed for both HS1 and HS2, with HS2 generally showing slightly slower decay times due to the slightly larger twist angle (see Supporting Note 2).

Fig. 3.7.1 was included in the revised version of our work as Supporting Note 5, and the following statement was added to the main text:

Power-dependent tr- μ -PL measurements on both HSs, reported in Supporting Note 4 and 5, show a progressive shortening of the time decay of the MX band as P_{exc} increases. This is a likely consequence of exciton-exciton interactions, that tend to diminish the exciton lifetime [24, 25], and of the increasing contribution to the MX band from de-trapped IXs with reduced lifetime, as discussed above.

3.8

"The different states of the moiré potential also present a different formation dynamics. Fig. 2c shows the time evolution of the MX signal up to 1 ns after the laser pulse excitation. In this case, the data are reproduced" It would be preferable to include the laser pulse in the reported data. As the detection is performed throughout the sample, the observed long lifetime could potentially be attributed to the time it takes for the detected exciton to flow within the sample, reach a defect or the edge of the sample, and recombine. Moreover, the influence of IX-IX repulsion has not been taken into account in this case, which could be a source of the observed phenomena. Conducting power-dependent lifetime measurements with a confocal setup could help clarify this ambiguity. As of now, the conclusion drawn from the data seems questionable.

We thank the Referee for the suggestion to include the laser pulse. As mentioned in the reply to the previous point, we have added a Supporting Figure (see Supporting Note 3) where we have included the laser pulse.

The Referee's hypothesis about the origin of the long time decay as due to a signal stemming from a region far away from the laser spot can be excluded thanks to the use of a confocal set-up as remarked in the previous point.

We cannot follow the Referee's argument about the fact that "*IX-IX repulsion...could be a source of the observed phenomena*", namely it seems here that the Referee implies that IX-IX interaction should lead to the "*observed long lifetime*". Instead, IX-IX interactions lead to a reduction of the exciton lifetime, as we observe in Fig. 4.2 c of Supporting Note 4 and reported, *e.g.*, in Refs. [24, 25].

3.9

"to $6.25 \cdot 10^{10} \text{ cm}^{-2}$ and a sizeable exciton-exciton interaction is possible thus explaining the decrease in the emission decay time observed in Fig. 2b as well as the MX band bluishift with P_{exc} " The assumption is that when the sample is excited, all the IX traps in the vicinity are also excited. One can view this as exciting the sample at point A, where some IX traps flow out of the light cone due to IX-IX interaction. The IX traps that flow away will eventually become trapped in different sites or defects throughout the sample. Therefore, the density per trap in this case should encompass the entire sample area rather than just the excitation area.

As already discussed above, we do not collect the signal from the entire sample area. We expect that in general with a high NA objective one does not collect the signal from the entire sample but just from a region close to the excitation area. This is particularly true in a confocal set-up as the one employed in our μ -PL experiments.

3.10

"photoluminescence (PL) spectra of the studied WSe₂/MoSe₂ heterostructure (HS)" The acronyms have already been defined previously, so there is no need to define them again.

We thank the Referee for noticing it, we have now corrected this part by using the acronyms.

3.11

"PL integrated intensity dependence on the laser power P_{exc} for MX (azure symbols) and X (dark yellow symbols) bands at $T=6 \text{ K}$ (full symbols) and $T=90 \text{ K}$ (open symbols). Solid and dashed lines are fits to the data with Eq. 3 for $T=6 \text{ K}$ and 90 K , respectively. At $T=6 \text{ K}$, the α coefficient values are 0.55 ± 0.02 and 0.89 ± 0.02 for MX and X, respectively. At $T=90 \text{ K}$, the α coefficient values are 0.99 ± 0.02 and 0.97 ± 0.03 for MX and X, respectively" Is the analysis method solely based on integrating the data, or does it involve fitting Lorentzian or Gaussian functions? It would be preferable to provide a more detailed explanation of the analysis method used to examine the data.

We thank the Referee for this question, we recognise this was not discussed in detail. Our analysis is actually not based on fitting due to the fact that (especially at low temperature) the shape of the PL bands changes when increasing power and at low powers in particular, where many narrow lines can be seen, a fitting would not be much accurate. We thus performed our analysis by considering a given energy range that includes the whole band and calculated the integrated intensity in that energy range.

We included the following statement in the main text when discussing Fig. 3:

The integrated intensity was obtained by performing integrals over suitable energy ranges (it cannot be reliably obtained by fitting the data due to the fact that the shape of the PL bands changes when increasing power and at low powers in particular, where many narrow lines can be seen, a fitting would not be accurate).

3.12

"Temperature variation of the α coefficient for the MX-IX and X bands. In the former case, a clear transition from a sublinear to a linear behaviour is found and ascribed to the transition from a moiré localisation regime to a free interlayer exciton one (hence the mixed label MX-IX)" It would be advisable to provide a more comprehensive and detailed explanation to support this claim. Furthermore, it would be beneficial to clarify if the power range for all the measurements is the same. If not, how do you ensure that the observed phenomena are not merely the low-power behavior of the system, which has not been recorded in higher temperatures.

We thank the Referee for this remark. The results shown in Fig. 3b of the manuscript were derived from the analysis reported in Supporting Note 4 of the previous version of the manuscript (now Supporting Note 6 for HS1 and Supporting Note 7 for HS2). The P_{exc} range was over than or equal to 3 orders of magnitude from $T=6$ K to 220 K and at least 2 orders of magnitude up to room temperature. However, it should be noted that already from $T \gtrsim 100$ K both the IX and the MX-IX bands exhibit a linear dependence of the integrated PL intensity at the same P_{exc} values for which, at lower T , the power law was sublinear, thus ruling out the conjecture of the Referee.

The Referee advises to provide a more comprehensive and detailed explanation to support our claims. We think that observing the same behaviour of the power law of the different exciton transitions in the second HS (HS2) reinforces our claims.

Indeed, the same phenomenology observed for HS1 was found also for HS2. As noted in Fig. 3d of the main text for HS1, at intermediate temperatures of about 90 K a dramatic lineshape change can be observed when increasing power, showing a clear transition from the MX prevailing at low P_{exc} to the IX prevailing at high P_{exc} . The same trend was indeed observed also for HS2, as demonstrated by the power-dependent spectra acquired at 80 K and shown in Fig. 3.12.1.

Figure 3.12.1: Left: $T=90$ K μ -PL spectra of HS1 for different laser excitation powers in the energy region where the MX and IX recombinations can be simultaneously observed. IX takes over MX upon increase of the photo-generated carrier density. Right: Same at $T=80$ K for HS2.

Power-dependent measurements on HS2 were performed at different temperatures between 6 K and 296 K, analogously to HS1. The integrated PL intensity of the MX/IX and of the X band

were then analysed and fitted by Eq. 3 of the main text, leading to the α coefficients shown in Fig. 3.12.2.

Figure 3.12.2: Temperature variation of the α coefficient for the MX-IX and X bands, comparison between HS1 and HS2. For HS1, some points measured before hBN-capping were also included in the plot, showing that the data agree well with those acquired after hBN-capping. Both for HS1 and HS2, a clear transition from a sublinear to a linear behaviour is found for the MX-IX band, which is ascribed to the transition from a moiré localisation regime to a free interlayer exciton one. Such a transition occurs at ≈ 120 K for HS1, and for ≈ 100 K for HS2, as highlighted by the black dashed lines.

In the plot concerning HS1, we also added some points which were measured before capping the sample with hBN (*i.e.*, on the bare HS on SiO_2). Indeed, these points follow the same trend of the data after hBN-capping, showing that substrate-related defects do not have an impact on the α vs T behaviour. In addition, thanks to the different twist angle θ of HS2, we find that the intensity crossover of the MX-IX transition from a sublinear to a linear behaviour depends on θ consistently with the shallower moiré potential for larger θ . This is now properly emphasized in the revised version of the manuscript. In particular, Figs. 1.1.2 and 1.1.3, along with the whole set of plots showing the dependence of the integrated PL intensity of the MX/IX and X bands on laser power, from which the α coefficients were derived, were included and discussed in Supporting Note 7. Furthermore, the following text was added in the main manuscript:

A similarly comprehensive study was performed also for HS2, which has a larger twist angle of 0.74° . The results are shown in Supporting Note 7. As for HS1, the X band shows a linear behavior ($\alpha=1$) all over the temperature range. Although also in HS2 the coefficient α for the MX-IX band exhibits a progressive increase from 0.5 to 1 (see Supporting Note 7), the plateau-value of 1 is reached at a temperature of about 100 K (lower than the 120 K value observed in HS1; see Fig. 3c) consistently with the shallower moiré potential at greater twist angles [4]. On the one hand, these results strengthen the picture described so far. On the other hand, they point to the crucial role of the twist angle in determining the thermal stability of moiré excitons and associated phenomena in moiré superlattices.

3.13

"T=90 K μ -PL spectra for different laser excitation powers in the energy region, where the MX and IX recombinations can be simultaneously observed. IX takes over MX upon increase of the photogenerated carrier density. e Same as d for T=296 K, where only the IX transition is observable." If the power is increased to the point where the trapped excitons become free, they should be able to gain momentum and move out of the light cone. In such a scenario, it raises the question of why we still observe them if they are completely free. Moreover, what prevents the free IX-IX from interacting with each other, leading to the disappearance of the signal?

The Referee raises a potentially interesting point concerning the "exit" of the interlayer excitons, IXs, out of the light cone. The Referee seems suggesting that, due to an increase of the photogenerated carrier density, excitons gain momentum to a point that they go out of the light cone in k-space, thus becoming not observable. However, the scenario envisaged by the Referee was not observed, for instance, in Ref. [15], where a WSe₂/MoSe₂ HS was studied by PL with photogenerated carrier densities as large as $n_{e-h}=3.2 \cdot 10^{14} \text{ cm}^{-2}$ without noticing a sizable decrease in the PL intensity, namely in the observation of the IXs. In our case, the largest n_{e-h} value reached was $=2 \cdot 10^{13} \text{ cm}^{-2}$. Furthermore, a recent theoretical work [5] showed that increasing the IX-IX interaction by increasing the exciton density gives rise to a blue shift of the PL peak of the moiré-related band. However, no disappearance of the exciton signal, whether related to moiré-confined interlayer excitons or to free interlayer excitons, was mentioned. While, we cannot exclude that exciton scattering out of the light cone takes place when the density of photogenerated excitons is increased, we have no compelling evidence of such phenomenon. Concerning the effect of temperature, in Ref. [3] (in particular, see Supplemental Figure S5 of that work) it was shown that a thermally broadened IX (either moiré-confined or not) distribution may even lead to a significantly increased exciton population within the light cone, the extent of such effect depending on the twist angle. In summary, we have no clear indications that temperature and/or exciton density increase cause IX moving out of the light cone. We would be grateful to the Referee if she/he could indicate some works reporting the phenomenology she/he is referring to.

3.14

"shown in Supporting Note 5. From T=6 K to T=120 K, the HS signal is dominated by the MX band, which undergoes a redistribution of the carrier population between the different states of the moiré potential. Starting from T=120 K, a high-energy component due to IXs appears and becomes increasingly important relative to the MX band," As the temperature is raised, energy is added to the system, causing the trapped excitons to move to higher energy levels. In such a scenario, it is puzzling that the exciton with lower energy level starts to exhibit higher intensity than the one with higher energy level. Additionally, the 80 meV energy difference between the MX and IX is not entirely trustworthy since the sample's inhomogeneity has not been measured or addressed.

As shown in Fig. 3.2.1 of this response, the PL spectra recorded at different locations of the HS are quite homogeneous. Therefore, the spectral properties observed and discussed can be considered as a general property of the sample.

The Referee makes a sensible observation concerning the relative exciton population of the moiré manifold at low T and with increasing the carrier thermal energy as shown in Fig. 4a of the main text and in Fig. 5.1 of the Supporting Note 5 of the previous version of the manuscript. Our findings coincide with those reported and discussed in Ref. [1], where the lowest energy resonance state does not exhibit the highest relative intensity between the moiré resonances. The authors of Ref. [1] ascribe this feature to disorder. Rather, we believe that the charge transfer among

different layers of the HS and valley relaxation dynamics make the relative occupancy of the exciton level quite complex that necessitates a specific series of experiments. Like in Ref. [1], we also observe that with increasing T the relative spectral weight of the highest-energy resonances decreases in favor of the lowest-energy resonances, as noted by the Referee. We believe this stems from the interplay between the carrier relaxation paths (possibly ruled by electronic selection rules; different exciton resonances correspond indeed to different angular momentum-like states [1]) and carrier thermal re-distribution between states under quasi-equilibrium conditions that are established by the external optical excitation. While we are aware this is a very qualitative explanation, we think that a dedicated and diverse series of experiments is necessary to account in detail for the thermal occupancy of the inter-excitonic levels in a moiré potential.

3.15

"the σ^+ red component is at higher energy than the σ^- blue one). Then, magneto- μ -PL measurements were performed also at $T = 160$ K ($P_{exc}=75 \mu W$), where the HS exciton band is instead dominated by free IXs. Fig. 4d shows the σ^+ and σ^- components of the IX spectra at different magnetic fields." At 160 K, there is a coexistence of both the well-defined MX and IX. To demonstrate that these two excitons have different origins that are independent of temperature and power, it is recommended to measure the g factor for both at the same temperature. Additionally, displaying the map of IX energy shift by the magnetic field, as seen in Figure 4b for the 160 K data, would be beneficial. Moreover, if the IX is playing a role in the low-temperature PL at high enough power, as claimed, then such a g factor should be observable.

We thank the Referee for this remark.

First of all, to ensure the general validity of our results, we performed similar measurements on HS2. Fig. 3.15.1a shows the polarisation-resolved magneto- μ -PL spectra acquired on HS2 at 10 K with $P_{exc} = 50$ nW. While the field increases, the energy of the narrow lines (see, *e.g.*, lines denoted as M1 and M2) is higher for the σ^+ polarisation with respect to the σ^- one, and the g -factor is thus positive. Fig. 3.15.1b shows instead the spectra acquired at 160 K with $P_{exc} = 100 \mu W$. In this case, the IX can be observed, and the σ^- component lies at higher energy than the σ^+ component, indicating a negative g -factor.

A quantitative analysis of the g -factor of the moiré narrow lines M1 and M2 at 10 K and of the IX band at 160 K is shown in Fig. 3.15.2. The g -factor values estimated for HS2 are indeed very close to those found for HS1 and displayed in Fig. 4d of the main text. Figs. 3.15.1 and 3.15.2 were included and discussed in Supporting Note 9.

To address the comment of the Referee, we also performed further studies on HS1. We recorded magneto- μ -PL spectra at various temperatures, comprising those where the MX and IX coexist. However, disentangling the contribution of the overlapping MX and IX bands in the spectrum is not feasible, especially considering the necessity of using high laser powers to attain a sizable PL signal. This inevitably leads to a prevailing signal from the free IX band. Nevertheless, we repeated high- T (80 K) and high-field (16 T) magneto- μ -PL measurements aimed at highlighting the MX component with respect to the free IX band. To do so, we employed very low laser powers of 10 nW. The results are shown in Fig. 3.15.3. We acquired the spectra at 0 T and 16 T (a whole sweep was not feasible because of the very low PL signal due to the low power and high T employed) and observed a clear splitting of the MX band at 16 T, from which we derive an estimate of the g -factor of about +6. The data agree with previous assignments demonstrating that the positive gyromagnetic factor for MX is indeed found also at 80 K. Fig. 3.15.3 was included in Supporting Note 11.

As for the suggestion of the Referee to measure the g -factor for both the MX and IX at the same temperature but different P_{exc} , this was already done in the previous version of the manuscript in Supporting Note 7, where it was shown how at 10 K and high enough power the

Figure 3.15.1: Magneto- μ -PL measurements on HS2 at low and high T s. **a** Polarisation-resolved magneto- μ -PL spectra acquired on HS2 with $P_{\text{exc}} = 50$ nW from 0 to 16 T in steps of 0.5 T. Two moiré narrow lines (Ms) are indicated. **b** Polarisation-resolved magneto- μ -PL spectra at $T = 160$ K and $P_{\text{exc}} = 100$ μ W of the free IX band for σ^+ and σ^- polarisations. A negative ZS can be observed, with the σ^+ and σ^- spectra being at lower and higher energy, respectively.

Figure 3.15.2: g -factor of the MXs and IX of HS2. **d** ZS of the two moiré-localised excitons M1 and M2 highlighted in Fig. 3.15.1a and of the free IX exciton shown in Fig. 3.15.1b vs magnetic field, resulting in the g -factors displayed in the figure. The ZS data of the M2 line was shifted by y-offset (by 2 meV) for ease of visualisation.

g -factor turns negative. That figure has now been moved to the main text (Figure 5 of the revised manuscript) to give it more emphasis in our work.

Concerning the Referee's suggestion to display the data in Fig. 4d of the previous version of the manuscript (magneto-PL data at 160 K and 75 μ W, now shown in Fig. 4c) as a colourplot rather than as stacked spectra, the main issue is that at high T the spectra are rather broad, so that displaying them as a colourplot does not allow one to appreciate well the opposite shift of the differently polarised components, namely the Zeeman splitting of the band. Following the Referee's comment, we however agree that it might be better to show both the low T and high

Figure 3.15.3: Polarisation-resolved μ -PL spectra acquired at 0 and 16 T with a power as low as 10 nW, in order to highlight the MX contribution. The low energy peak, attributed to the MX, shows a clear positive Zeeman splitting. At each field, the MX peak of the σ^- component is normalised to 1.

T data with a similar style. We thus displayed the low T data as stacked spectra rather than as colourplots. Furthermore, following the requests of Referee 2, we acquired new measurements at low T on HS1 with powers as low as 10 nW. We thus modified figure 4 by including these data instead of those previously shown.

As for the last statement of the Referee, claiming that if the IX plays a role at low T then its negative g -factor should be observable, it is certainly so. This was already demonstrated by the data in Supporting Note 7 of the previous version of the manuscript (now Figure 5 of the revised manuscript). This is further supported by our new magneto-PL measurements performed as a function of power at 16 T, showing how the g -factor turns from positive to negative when increasing laser power. Such measurements are shown in Fig. 3.15.4 for both HS1 and HS2 (we apologise to show these data once more, but we did it for ease of reading to avoid jumping from a point to the other of the response). Indeed, at low power a positive splitting of the narrow lines can be observed (see, *e.g.*, the narrow lines highlighted the black circles). By increasing power, a high energy component emerges, and the g -factor globally becomes negative. This can be noticed especially for HS1. For HS2, the g -factor at high power is negative but small, presumably due to a coexistence of localised states and free excitons at the largest power we used.

3.16

"At $T=100$ K, τ_r approaches the temporal resolution limit. The higher temperatures and the ensuing MX ionisation process result indeed in a decreased contribution of the moiré localisation step and thus in a reduction of the time required to build up the exciton population contributing to the MX/IX band—" Can it can simply be explained through faster flow of IX?

The hypothesis of the Referee could be plausible. However, we believe we provided numerous evidences that a MX to free IX transition takes place. Such evidences are based on diverse and complementary measurements, which comprise time-resolved and steady-state μ -PL with a confocal set-up also under high magnetic fields up to 16 T and varying the sample temperature from 6 K to room temperature and the photo-excited carrier density over nearly four orders of magnitude. In addition and thanks to the Referees' comments, we further reinforced our picture

Figure 3.15.4: Polarisation-resolved μ -PL power studies performed at 10 K under a 16 T magnetic field for both HS1 and HS2. At low powers, several Zeeman-split narrow lines can be observed, such as those highlighted by the black circle. At high powers, a change in the sign of the Zeeman splitting can clearly be noticed.

by considering two HSs with different twist angles (*i.e.*, different moiré potential depth) that provide a clear trend and thus reinforce our picture.

3.17

In the method section the wavelength of the excitation for measuring the magnetic field PL is not clarified.

We respectfully notice that the excitation wavelength was actually already specified in the Methods. The following sentence was in fact therein reported: "A 515-nm-laser and a 100 \times microscope objective with NA=0.82 were used."

Finally, we thank the Referee once more for her/his numerous comments, which we believe that allowed us to improve our work.

References

- [1] K. Tran, G. Moody, F. Wu, X. Lu, J. Choi, K. Kim, A. Rai, D. A. Sanchez, J. Quan, A. Singh, J. Embley, A. Zepeda, M. Campbell, T. Autry, T. Taniguchi, K. Watanabe, N. Lu, S. K. Banerjee, K. L. Silverman, S. Kim, E. Tutuc, L. Yang, A. H. MacDonald, and X. Li, *Evidence for moiré excitons in van der Waals heterostructures*, Nature **567**, 7746, 71–75 (2019).
- [2] Z. Li, X. Lu, D. F. C. Leon, Z. Lyu, H. Xie, J. Hou, Y. Lu, X. Guo, A. Kaczmarek, T. Taniguchi, K. Watanabe, L. Zhao, L. Yang, and P. B. Deotare, *Interlayer Exciton Transport in MoSe₂/WSe₂ Heterostructures*, ACS Nano **15**, 1539 (2021).
- [3] J. Choi, M. Florian, A. Steinhoff, D. Erben, K. Tran, D. S. Kim, L. Sun, J. Quan, R. Claassen, S. Majumder, J. A. Hollingsworth, T. Taniguchi, K. Watanabe, K. Ueno, A. Singh, G. Moody, F. Jahnke, and X. Li, *Twist Angle-Dependent Interlayer Exciton Lifetimes in van der Waals Heterostructures*, Phys. Rev. Lett. **126**, 047401 (2021).
- [4] B.-H. Lin, Y.-C. Chao, I. Hsieh, C.-P. Chuu, C.-J. Lee, F.-H. Chu, L.-S. Lu, W.-T. Hsu, C.-W. Pao, C.-K. Shih, J.-J. Su, , and W.-H. Chang, *Remarkably Deep Moiré Potential for Intralayer Excitons in MoSe₂/MoS₂ Twisted Heterobilayers*, Nano Lett. **23**, 1306 (2023).
- [5] S. Brem and E. Malic, *Bosonic Delocalization of Dipolar Moiré Excitons*, Nano Lett. **23**, 4627 (2023).
- [6] T. Woźniak, P. E. F. Junior, G. Seifert, A. Chaves, and J. Kunstmann, *Exciton g factors of van der Waals heterostructures from first-principles calculations*, Phys. Rev. B **100**, 235408 (2020).
- [7] M. Brotons-Gisbert, H. Baek, A. Campbell, K. Watanabe, T. Taniguchi, and B. D. Gerardot, *Moiré-Trapped Interlayer Trions in a Charge-Tunable WSe₂/MoSe₂ Heterobilayer*, Phys. Rev. X **11**, 031033 (2021).
- [8] F. MahdikhanySarvejahany, D. N. Shanks, M. Klein, Q. Wang, M. R. Koehler, D. G. Mandrus, T. Taniguchi, K. Watanabe, O. L. Monti, B. J. LeRoy, and J. R. Schaibley, *Localized Interlayer Excitons in MoSe₂/WSe₂ Heterostructures without a Moiré Potential*, Nat. Commun. **13**, 5354 (2022).
- [9] E. Liu, E. Barré, J. van Baren, M. Wilson, T. Taniguchi, K. Watanabe, Y.-T. Cui, N. M. Gabor, T. F. Heinz, Y.-C. Chang, and C. H. Lui, *Signatures of moiré trions in MoSe₂/WSe₂ heterobilayers*, Nature **594**, 46 (2021).
- [10] M. R. Rosenberger, H.-J. Chuang, M. Phillips, V. P. Oleshko, K. M. McCreary, S. V. Sivaram, C. S. Hellberg, and B. T. Jonker, *Twist Angle-Dependent Atomic Reconstruction and Moiré' Patterns in Transition Metal Dichalcogenide Heterostructures*, ACS Nano **14**, 4550 (2020).
- [11] T. Wang, S. Miao, Z. Li, Y. Meng, Z. Lu, Z. Lian, M. Blei, T. Taniguchi, K. Watanabe, S. Tongay, D. Smirnov, and S.-F. Shi, *Giant Valley-Zeeman Splitting from Spin-Singlet and Spin-Triplet Interlayer Excitons in WSe₂/MoSe₂ Heterostructure*, Nano Lett. **20**, 694 (2020).
- [12] F. MahdikhanySarvejahany, D. N. Shanks, C. Muccianti, B. H. Badada, I. Idi, A. Alfrey, S. Raglow, M. R. Koehler, D. G. Mandrus, T. Taniguchi, K. Watanabe, O. L. A. Monti, H. Yu, B. J. LeRoy, and J. R. Schaibley, *Temperature dependent moiré trapping of interlayer excitons in MoSe₂-WSe₂ heterostructures*, npj 2D Mater. Appl. **5**, 67 (2021).

- [13] H. Kim, D. Dong, Y. Okamura, K. Shinokita, K. Watanabe, T. Taniguchi, and K. Matsuda, *Dynamics of Moiré Trion and Its Valley Polarization in a Microfabricated $WSe_2/MoSe_2$ Heterobilayer*, ACS Nano **17**, 13715 (2023).
- [14] B. Miller, A. Steinhoff, B. Pano, J. Klein, F. Jahnke, A. Holleitner, and U. Wurstbauer, *Long-Lived Direct and Indirect Interlayer Excitons in van der Waals Heterostructures*, Nano Lett. **17**, 5229 (2017).
- [15] J. Wang, J. Ardelean, Y. Bai, A. Steinhoff, M. Florian, F. Jahnke, X. Xu, M. Kira, J. Hone, and X.-Y. Zhu, *Optical generation of high carrier densities in 2D semiconductor heterobilayers*, Sci. Adv. **5**, eaax0145 (2019).
- [16] W. Li, X. Lu, J. Wu, and A. Srivastava, *Optical control of the valley Zeeman effect through many-exciton interactions*, Nat. Nanotechnol. **16**, 148 (2021).
- [17] K. L. Seyler, P. Rivera, H. Yu, N. P. Wilson, E. L. Ray, D. G. Mandrus, J. Yan, W. Yao, and X. Xu, *Signatures of moiré-trapped valley excitons in $MoSe_2/WSe_2$ heterobilayers*, Nature **567**, 66 (2019).
- [18] W. Li, X. Lu, S. Dubey, L. Devenica, and A. Srivastava, *Dipolar interactions between localized interlayer excitons in van der Waals heterostructures*, Nat. Mater. **19**, 624 (2020).
- [19] H. Baek, M. Brotons-Gisbert, Z. X. Koong, A. Campbell, M. Rambach, K. Watanabe, T. Taniguchi, and B. D. Gerardot, *Highly energy-tunable quantum light from moiré-trapped excitons*, Sci. Adv. **6**, 37, eaba8526 (2020).
- [20] M. Troue, J. Figueiredo, L. Sigl, C. Paspalides, M. Katzer, T. Taniguchi, K. Watanabe, M. Selig, A. Knorr, U. Wurstbauer, and A. W. Holleitner, *Extended Spatial Coherence of Interlayer Excitons in $MoSe_2/WSe_2$ Heterobilayers*, Phys. Rev. Lett. **131**, 036902 (2023).
- [21] P. Parzefall, J. Holler, M. Scheuck, A. Beer, K.-Q. Lin, B. Peng, B. Monserrat, P. Nagler, M. Kempf, and T. Korn, *Moiré phonons in twisted $MoSe_2-WSe_2$ heterobilayers and their correlation with interlayer excitons*, 2D Mater. **8**, 035030 (2021).
- [22] H. Kim, K. Aino, K. Shinokita, W. Zhang, K. Watanabe, T. Taniguchi, and K. Matsuda, *Dynamics of Moiré Exciton in a Twisted $MoSe_2/WSe_2$ Heterobilayer*, Adv. Optical Mater. **11**, 2300146 (2023).
- [23] H. Guo, X. Zhang, and G. Lu, *Moiré excitons in defective van der Waals heterostructures*, Proc. Natl. Acad. Sci. U. S. A. **118**, e2105468118 (2021).
- [24] G. Aivazian, H. Yu, S. Wu, J. Yan, D. G. Mandrus, D. Cobden, W. Yao, and X. Xu, *Many-body effects in nonlinear optical responses of 2D layered semiconductors*, 2D Mater. **4**, 025024 (2017).
- [25] O. Salehzadeh, N. H. Tran, X. Liu, I. Shih, and Z. Mi, *Exciton Kinetics, Quantum Efficiency, and Efficiency Droop of Monolayer MoS_2 Light-Emitting Devices*, Nano Lett. **14**, 4125 (2014).

Reviewers' Comments:

Reviewer #1:

Remarks to the Author:

The revised manuscript has fully addressed the reviewers' comments. I would like to recommend the acceptance without any changes.

Reviewer #2:

Remarks to the Author:

I think the authors have done a great work on addressing my concerns and improving the manuscript during revisions. The conclusion reached by the paper are now scientifically sound and clear. Therefore, I recommend the publication of this paper on Nature Communications now.

Reviewer #3:

Remarks to the Author:

I thank the authors for their comprehensive responses to the comments. The redoing of measurements on a new device and the observation of a consistent trend across multiple devices are reassuring, especially in systems known for variability. At this point, I have a few minor comments and suggestions but no major concerns.

Regarding the device type, I am curious if constructing an H-type device would be interesting. Such systems, often characterized by a shallow potential well, seem easier to study at lower temperatures with reduced power and magnetic fields. As you might know, in H-type systems, increasing the power can lead to the emergence of a new peak with a different g-factor. It might be worth checking "Giant Valley-Zeeman Splitting from Spin-Singlet and Spin-Triplet Interlayer Excitons in WSe₂/MoSe₂ Heterostructure" by Tianmeng Wang, et al., and citing their work.

For further insights, qualitatively, there is no notable change between the spectral shapes of neutral and charged IX at small doping densities (see "Electrical control of interlayer exciton dynamics in atomically thin heterostructures" by Jauregui, L. A., et al. (2019)). Therefore, the authors cannot refer to the shape as a sign of quality. The center of energy shifting to lower energies in the silicon-based sample, compared to the hBN-encapsulated sample (as seen in figure 3.1.2), could be a signature of doping and indicative of lower quality samples. The role of hBN in enhancing quality by eliminating screening effects and avoiding unwanted charging in the system has been studied in the well-known work of Alexey Chernikov, et al. Phys. Rev. Lett. 113, 076802 (2014), led by Professor Tony Heinz.

Regarding your mention of "The smaller g_{exc} for the Gaussian components might be caused by exciton-exciton interactions," this is not widely known information. Adding a sentence or two to explain or a citation would be beneficial. Additionally, the sign of the g-factor might be influenced by the dominant P matrix element at the site with the dominant emissions in the heterostructure, as discussed by Dr. Hongyi Yu in "Moiré excitons: From programmable quantum emitter arrays to spin-orbit-coupled artificial lattices." Also, Tomasz Woźniak, et al., in "Exciton g factors of van der Waals heterostructures from first-principles calculations" explain the origin of the g-factor in heterostructures.

As you know, any localizing factors such as moiré patterns, strain, nano bubbles created during the transfer process, and defects can lead to the observation of narrow lines in both samples. Hence, the observation of narrow lines does not necessarily reflect the quality of the devices. Additionally, the roughness of the SiO₂ wafer is about 2nm. Considering the size of nanoparticles used for creating single quantum emitters in these systems is in the order of 5 to 50 nm, this showcases the effect of such roughness and another reason why encapsulating your heterostructures is a better practice.

By increasing the temperature, phonon-mediated recombination occurs. In heterostructures with

even larger twist angles, phonons can assist in the recombination of electrons and holes at higher temperatures, indicating that the observation of this phenomenon is not exclusively a hallmark of device quality in terms of small twist angles. Since the sample is not gated to change the doping and observe the small neutral region, the best approach might be to emphasize the homogeneity of the sample and the small and comparable energy range of the interlayer exciton PL emission. In perfect alignment, when exciting the system, you should have a fast non-radiative decay between the conduction and valence bands of MoSe₂ and WSe₂, and your prominent radiative decay should be your IX since it is energetically favorable until your IX signal gets saturated. As the twist angle increases and you deviate from the perfect alignment of the K points of the two layers, you observe an increase in monolayer PL emissions and less quenching.

While fitting and estimating the twist angle of heterostructures is a known procedure, as used in "Evidence for moiré excitons in van der Waals heterostructures" by Khan Tran, et al., considering the uncertainty in fitting, it is good practice to exercise caution and clarity in claims made with these estimations. In systems where SHG cannot be measured with reasonable uncertainty, it is better to perform PFM on the sample before capping it with hBN and, based on the heterostructure lattice constant, obtain an accurate twist angle.

Regarding Supplementary Note 12, you mention that the peak appearing around 1.4 eV is associated with free interlayer excitons (IX). Notably, in Figure 12.1 of the supporting notes, there is a peak near this energy in HS2 (this peak is more prominent for HS2 compare to HS1), which might also be attributed to free IX. Consequently, one might expect a more pronounced negative Zeeman shift in HS2 compared to HS1. However, this is not readily apparent to the naked eye in your figure. I suggest acknowledging this observation in your discussion.

Point-by-point reply to the Referees

Localisation-to-delocalisation transition of moiré excitons in WSe₂/MoSe₂ heterostructures

Elena Blundo,^{1,*} Federico Tuzi,¹ Salvatore Cianci,¹ Marzia Cuccu,¹ Katarzyna Olkowska-Pucko,² Łucja KipczaK,² Giorgio Contestabile,¹ Antonio Miriametro,¹ Marco Felici,¹ Giorgio Pettinari,³ Takashi Taniguchi,⁴ Kenji Watanabe,⁵ Adam Babiński,² Maciej R. Molas,² and Antonio Polimeni^{1,*}

¹ *Physics Department, Sapienza University of Rome, 00185, Roma, Italy.*

² *Institute of Experimental Physics, Faculty of Physics, University of Warsaw, Pasteura 5, 02-093 Warsaw, Poland*

³ *Institute for Photonics and Nanotechnologies (CNR-IFN), National Research Council, 00133, Rome, Italy*

⁴ *International Center for Materials Nanoarchitectonics, National Institute for Materials Science, 1-1 Namiki, Tsukuba 305-0044, Japan.*

⁵ *Research Center for Functional Materials, National Institute for Materials Science, 1-1 Namiki, Tsukuba 305-0044, Japan*

* Corresponding authors:

elena.blundo@uniroma1.it,

antonio.polimeni@uniroma1.it

Contents

Reply to Referee 1	1
Reply to Referee 2	2
Reply to Referee 3	3
3.1	3
3.2	3
3.3	3
3.4	4
3.5	4
3.6	5
3.7	5
References	7

Referee 1

The revised manuscript has fully addressed the reviewers' comments. I would like to recommend the acceptance without any changes.

We thank the Referee for her/his positive feedback on our manuscript and for recommending it for publication.

We would like also to let the Referee know that we have included a more in-depth discussion concerning the interpretation of the results shown in Fig. 5. After we posted our manuscript on arXiv, we received a comment about the data displayed in Fig. 5b showing magneto- μ -PL measurements performed at low temperature and large photo-excited carrier density. In the previously revised version of the manuscript, we attributed the negative sign of the gyromagnetic factor of the higher energy states of the MX band to the prevailing contribution of free interlayer excitons at high laser power density (namely, high photo-excited carrier density). However, we should also consider the possibility that at elevated photo-excited carrier densities excitons may occupy R_h^h regions of the moiré potential once the R_h^X minima (*i.e.*, those prevailing at low temperature and not elevated photo-generated carrier densities) are saturated. In $WSe_2/MoSe_2$ HSs, the energy distance between these exciton states localised in minima with different local registry (*i.e.*, R_h^h and R_h^X) ranges from 70-80 meV [1, 2, 3] to 40 meV [4], but most importantly the lowest-energy exciton at R_h^X features a *positive* gyromagnetic factor, while the exciton at R_h^h is characterised by a *negative* gyromagnetic factor. More specifically, according to Ref. [5], $g_{exc,R_h^X} = +6.19$ and $g_{exc,R_h^h} = -6.15$. Remarkably, the latter value is very close to the value we found in Fig. 5c for the gyromagnetic factor $g_{exc,MX/IX} = -6.93$ of the MX component prevailing at $T=6$ K and $P_{exc} = 75 \mu W$. Eventually, at high temperatures ($T = 160$ K) the gyromagnetic factor of the HS exciton $g_{exc,IX} = -4.64$ (see Fig. 4d) is close to the value evaluated for a free IX (see Eq. 5 of the main text).

Referee 2

I think the authors have done a great work on addressing my concerns and improving the manuscript during revisions. The conclusion reached by the paper are now scientifically sound and clear. Therefore, I recommend the publication of this paper on Nature Communications now.

We thank the Referee for recognising our work and for recommending our manuscript for publication.

We would like also to let the Referee know that we have included a more in-depth discussion concerning the interpretation of the results shown in Fig. 5. After we posted our manuscript on arXiv, we received a comment about the data displayed in Fig. 5b showing magneto- μ -PL measurements performed at low temperature and large photo-excited carrier density. In the previously revised version of the manuscript, we attributed the negative sign of the gyromagnetic factor of the higher energy states of the MX band to the prevailing contribution of free interlayer excitons at high laser power density (namely, high photo-excited carrier density). However, we should also consider the possibility that at elevated photo-excited carrier densities excitons may occupy R_h^h regions of the moiré potential once the R_h^X minima (*i.e.*, those prevailing at low temperature and not elevated photo-generated carrier densities) are saturated. In $WSe_2/MoSe_2$ HSs, the energy distance between these exciton states localised in minima with different local registry (*i.e.*, R_h^h and R_h^X) ranges from 70-80 meV [1, 2, 3] to 40 meV [4], but most importantly the lowest-energy exciton at R_h^X features a *positive* gyromagnetic factor, while the exciton at R_h^h is characterised by a *negative* gyromagnetic factor. More specifically, according to Ref. [5], $g_{exc,R_h^X} = +6.19$ and $g_{exc,R_h^h} = -6.15$. Remarkably, the latter value is very close to the value we found in Fig. 5c for the gyromagnetic factor $g_{exc,MX/IX} = -6.93$ of the MX component prevailing at $T=6$ K and $P_{exc} = 75 \mu W$. Eventually, at high temperatures ($T = 160$ K) the gyromagnetic factor of the HS exciton $g_{exc,IX} = -4.64$ (see Fig. 4d) is close to the value evaluated for a free IX (see Eq. 5 of the main text).

Referee 3

I thank the authors for their comprehensive responses to the comments. The redoing of measurements on a new device and the observation of a consistent trend across multiple devices are reassuring, especially in systems known for variability. At this point, I have a few minor comments and suggestions but no major concerns.

We thank the Referee for her/his feedback. We also thank the Referee for her/his further comments and useful suggestions of pertinent papers. We address her/his questions below.

3.1

Regarding the device type, I am curious if constructing an H-type device would be interesting. Such systems, often characterized by a shallow potential well, seem easier to study at lower temperatures with reduced power and magnetic fields. As you might know, in H-type systems, increasing the power can lead to the emergence of a new peak with a different g-factor. It might be worth checking "Giant Valley-Zeeman Splitting from Spin-Singlet and Spin-Triplet Interlayer Excitons in WSe₂/MoSe₂ Heterostructure" by Tianmeng Wang, et al., and citing their work.

Certainly, H-type heterostructures (HSs) would be interesting due to their shallower potential as compared to that of R-type HSs. We expect that the localised-to-delocalised transition of moiré excitons would take place at lower temperature values. Though, it is hard to say if this circumstance makes it easier to study the moiré exciton de-trapping process. Possibly, the larger gyromagnetic factor featured by singlet and triplet moiré excitons in H-type HSs require less intense magnetic fields to observe changes in this quantity induced by carrier concentration density and temperature variations. We are aware of the paper by T. Wang *et al.*. Indeed, it was already included in the reference list of the first version of the manuscript as Ref. [25] (now as Ref. [24] in the last version of the manuscript).

3.2

For further insights, qualitatively, there is no notable change between the spectral shapes of neutral and charged IX at small doping densities (see "Electrical control of interlayer exciton dynamics in atomically thin heterostructures" by Jauregui, L. A., et al. (2019)). Therefore, the authors cannot refer to the shape as a sign of quality. The center of energy shifting to lower energies in the silicon-based sample, compared to the hBN-encapsulated sample (as seen in figure 3.1.2), could be a signature of doping and indicative of lower quality samples. The role of hBN in enhancing quality by eliminating screening effects and avoiding unwanted charging in the system has been studied in the well-known work of Alexey Chernikov, et al. Phys. Rev. Lett. 113, 076802 (2014), led by Professor Tony Heinz.

We agree that h-BN capping is beneficial for the fabrication of HSs and we now fabricate such structures on h-BN routinely. Based on the paper mentioned by the Referee, we can safely assume that the doping level of our HS1 should be small. The observation of the Referee about the energy shift of the moiré exciton after the capping of the sample is interesting and it indicates the likely relevant role of h-BN capping when the HS is fabricated on SiO₂.

3.3

Regarding your mention of "The smaller g_{exc} for the Gaussian components might be caused by exciton-exciton interactions," this is not widely known information. Adding a sentence or two to explain or a citation would be beneficial. Additionally, the sign of the g-factor might be influenced

by the dominant P matrix element at the site with the dominant emissions in the heterostructure, as discussed by Dr. Hongyi Yu in "Moiré excitons: From programmable quantum emitter arrays to spin-orbit-coupled artificial lattices." Also, Tomasz Woźniak, et al., in "Exciton g factors of van der Waals heterostructures from first-principles calculations" explain the origin of the g-factor in heterostructures.

Indeed, the analysis of the Zeeman splitting of the different gaussian components of the moiré manifold is not "widely known information" and it has been reported for the first time in our work. We hypothesise exciton-exciton interactions as a possible origin of the smaller gyromagnetic factor observed for the gaussian components with respect to the moiré single lines. A dedicated theoretical study taking into account many-body interaction effects on the electronic properties of moiré excitons would be necessary for explaining our finding. In that respect, our experimental observations provide clear indication that the moiré gyromagnetic factor of each gaussian component is independent of the moiré potential manifold energy (see Fig. 10.1 of the Supporting Information) and thus should be related to intracell interactions or carrier-induced screening of the moiré potential. Yet fascinating topics, they are beyond the scope of our work. We agree that the sign (and value) of the gyromagnetic factor is dictated by the atomic registry of the region of the HS, where the exciton recombination takes place. It is indeed on this ground that we modified the interpretation of part of the results presented in Fig. 5 of the manuscript (see point 3.7).

3.4

As you know, any localizing factors such as moiré patterns, strain, nano bubbles created during the transfer process, and defects can lead to the observation of narrow lines in both samples. Hence, the observation of narrow lines does not necessarily reflect the quality of the devices. Additionally, the roughness of the SiO2 wafer is about 2nm. Considering the size of nanoparticles used for creating single quantum emitters in these systems is in the order of 5 to 50 nm, this showcases the effect of such roughness and another reason why encapsulating your heterostructures is a better practice.

We agree that using a h-BN substrate is the optimal practise for HS fabrication and that several factors may contribute to the appearance of narrow lines in TMD monolayers and HSs. The origin of the narrow lines in HSs is indeed a quite hot topic. In our work, the facts that we deem relevant are the energy of those lines being resonant with the moiré exciton manifold and the Zeeman splitting of those lines being in agreement with the gyromagnetic factor expected for the pertinent moiré exciton states.

3.5

By increasing the temperature, phonon-mediated recombination occurs. In heterostructures with even larger twist angles, phonons can assist in the recombination of electrons and holes at higher temperatures, indicating that the observation of this phenomenon is not exclusively a hallmark of device quality in terms of small twist angles. Since the sample is not gated to change the doping and observe the small neutral region, the best approach might be to emphasize the homogeneity of the sample and the small and comparable energy range of the interlayer exciton PL emission. In perfect alignment, when exciting the system, you should have a fast non-radiative decay between the conduction and valence bands of MoSe2 and WSe2, and your prominent radiative decay should be your IX since it is energetically favorable until your IX signal gets saturated. As the twist angle increases and you deviate from the perfect alignment of the K points of the two layers, you observe an increase in monolayer PL emissions and less quenching.

The Referee’s analysis is correct. Nevertheless, we believe that it is rather reasonable to expect a larger radiative efficiency when k-space matching is attained. This should also lead to an overall larger radiative efficiency and, thus, to a higher probability to observe radiative recombination at high temperatures.

3.6

While fitting and estimating the twist angle of heterostructures is a known procedure, as used in "Evidence for moiré excitons in van der Waals heterostructures" by Khan Tran, et al., considering the uncertainty in fitting, it is good practice to exercise caution and clarity in claims made with these estimations. In systems where SHG cannot be measured with reasonable uncertainty, it is better to perform PFM on the sample before capping it with hBN and, based on the heterostructure lattice constant, obtain an accurate twist angle.

We are aware of this problem and we provided the uncertainty on the twist angle values accordingly. We guess that the Referee refers to piezoresponse force microscopy (PFM) as a valuable tool for determining the twist angle after the HS fabrication (but before h-BN capping). Of course, as for all scanning probe techniques, data processing is a rather important step and also in this case a direct twist angle measure is not straightforward. We agree, anyway, that PFM can provide useful information to complement the information obtained by SHG and the fitting analysis of the PL signal. We will keep it into consideration for future studies on 2D HSs.

3.7

Regarding Supplementary Note 12, you mention that the peak appearing around 1.4 eV is associated with free interlayer excitons (IX). Notably, in Figure 12.1 of the supporting notes, there is a peak near this energy in HS2 (this peak is more prominent for HS2 compare to HS1), which might also be attributed to free IX. Consequently, one might expect a more pronounced negative Zeeman shift in HS2 compared to HS1. However, this is not readily apparent to the naked eye in your figure. I suggest acknowledging this observation in your discussion.

In principle, if we deal with a free interlayer exciton (*i.e.*, moiré-detrapped exciton), its gyromagnetic should be similar independently of the HS twist angle and thus we would expect a similar ZS should be observed for both HS1 and HS2. This is not easily observable in Fig. 12.1 due to lineshape broadening effects induced by the large laser power densities employed. These densities may affect the emission of HS1 and HS2 to a different extent in a such a way that a direct comparison is not feasible.

This point raised by the Referee pairs a comment that we received after the first review round (after we posted our manuscript on arXiv) and that we now briefly discuss. The data displayed in Fig. 5b show magneto- μ -PL measurements performed at low temperature and large photo-excited carrier density. In the revised version of the manuscript, we attributed the negative sign of the gyromagnetic factor of the higher energy states of the MX band to the prevailing contribution of free interlayer excitons at high laser power density (namely, high photo-excited carrier density). However, we should also consider the possibility that at elevated photo-excited carrier densities excitons may occupy R_h^h regions of the moiré potential once the R_h^X minima (*i.e.*, those prevailing at low temperature and not elevated photo-generated carrier densities) are saturated. In WSe₂/MoSe₂ HSs, the energy distance between these exciton states localised in minima with different local registry (*i.e.*, R_h^h and R_h^X) ranges from 70-80 meV [1, 2, 3] to 40 meV [4], but most importantly the lowest-energy exciton at R_h^X features a *positive* gyromagnetic factor, while the exciton at R_h^h is characterised by a *negative* gyromagnetic factor. More specifically, according to Ref. [5], $g_{exc,R_h^X} = +6.19$ and $g_{exc,R_h^h} = -6.15$. Remarkably, the latter value is very close to the value we found in Fig. 5c for the gyromagnetic factor $g_{exc,MX/IX} = -6.93$ of

the MX component prevailing at $T=6$ K and $P_{\text{exc}} = 75 \mu\text{W}$. Eventually, at high temperatures ($T = 160$ K) the gyromagnetic factor of the HS exciton $g_{\text{exc,IX}} = -4.64$ (see Fig. 4d) is close to the value evaluated for a free IX (see Eq. 5 of the main text).

We have included this discussion when describing Fig. 5, which has been modified accordingly to show the photo-excited carrier density- and temperature-driven evolution of the exciton gyromagnetic factor.

Finally, we thank the Referee once more for her/his numerous and stimulating comments, which we believe allowed us to improve our work.

References

- [1] K. Tran, G. Moody, F. Wu, X. Lu, J. Choi, K. Kim, A. Rai, D. A. Sanchez, J. Quan, A. Singh, J. Embley, A. Zepeda, M. Campbell, T. Autry, T. Taniguchi, K. Watanabe, N. Lu, S. K. Banerjee, K. L. Silverman, S. Kim, E. Tutuc, L. Yang, A. H. MacDonald, and X. Li, *Evidence for moiré excitons in van der Waals heterostructures*, *Nature* **567**, 71 (2019).
- [2] Z. Li, X. Lu, D. F. C. Leon, Z. Lyu, H. Xie, J. Hou, Y. Lu, X. Guo, A. Kaczmarek, T. Taniguchi, K. Watanabe, L. Zhao, L. Yang, and P. B. Deotare, *Interlayer Exciton Transport in MoSe₂/WSe₂ Heterostructures*, *ACS Nano* **15**, 1539 (2021).
- [3] X. Lu and L. Yang, *Modulated interlayer exciton properties in a two-dimensional moiré crystal*, *Phys. Rev. B* **100**, 155416 (2019).
- [4] H. Yu, G.-B. Liu, J. Tang, X. Xu, and W. Yao, *Moiré excitons: From programmable quantum emitter arrays to spin-orbit-coupled artificial lattices*, *Sci. Adv.* **3**, e1701696 (2017).
- [5] T. Woźniak, P. E. F. Junior, G. Seifert, A. Chaves, and J. Kunstmann, *Exciton g factors of van der Waals heterostructures from first-principles calculations*, *Phys. Rev. B* **100**, 235408 (2020).